# Personalized Federated Learning on Flowing Data Heterogeneity under Restricted Storage

## Abstract

Recent years, researchers focused on personalized federated learning (pFL) to address the inconsistent requirements of clients causing by data heterogeneity in federated learning (FL). However, existing pFL methods typically assume that local data distribution remains unchanged during FL training, the changing data distribution in actual heterogeneous data scenarios can affect model convergence rate and reduce model performance. In this paper, we focus on solving the pFL problem under the situation where data flows through each client like a flowing stream which called Flowing Data Heterogeneity under Restricted Storage, and shift the training goal to the comprehensive performance of the model throughout the FL training process. Therefore, based on the idea of category decoupling, we design a local data distribution reconstruction scheme and a related generator architecture to reduce the error of the controllable replayed data distribution, then propose our pFL framework, pFedGRP, to achieve knowledge transfer and personalized aggregation. Comprehensive experiments on five datasets with multiple settings show the superiority of pFedGRP over eight baseline methods.

## 1 Introduction

Federated Learning (FL) (McMahan et al. (2017)) is an emerging distributed machine learning framework with privacy protection. In FL, the clients upload the locally trained model to the server for aggregation to reduce communication bandwidth and real-time requirements while avoiding direct exposure of potential sensitive data on the client, and the server aggregates the local models into a global model and distributes it to each client. However, in real-world applications, the data distribution within client and between clients varies over time(Li et al. (2020a)), and the accessible data on the client side is often limited by storage space and relevant regulations and policies(Voigt & Bussche (2017), Vizitiu et al. (2019)). For example, in the context of the COVID-19 pandemic, health institutions in different regions can use FL to conduct research while protecting data privacy(Yang et al. (2020)), but the high mutation rate of the virus can lead to differences in the distribution and trends of medical data across institutions (see Figure 1), and the original medical data usually cannot be stored for a long time in medical institutions(Voigt & Bussche (2017)), meaning that FL methods need to have strong robustness to be applied in such practical situation. We call the FL situation where data flows like a stream on each client as "Flowing Data Heterogeneity under Restricted Storage". Since the existence of a single global model can applicable to all clients is at odds with the

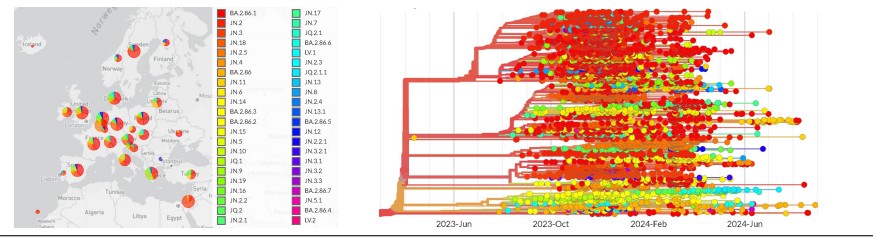

Figure 1: The proportion of virus types prevalent in various regions of Europe in July 2024, and the variation of COVID-19 BA.2.86 strain in various parts of Europe from August 2023 to July 2024. The data is sourced from `https://gisaid.org/hcov19-variants/`

fact of the statistical heterogeneity of data observed between different clients(Sattler et al. (2020), Kairouz et al. (2021)), FL methods should provide personalized global models for each client when data heterogeneity is unknown, which is also known as personalized Federated Learning (pFL).

Personalized Federated Learning methods improve the performance of the global models on the client side by trade-off the individual utility and collaborative benefits. Specifically, Chatterjee (2020) found that the similar small-batch gradients can improve model generalization and accelerate model convergence in machine learning, then Li et al. (2023) validated the conclusion above in FL setting and found that the similarity of local gradients are inversely proportional to the data heterogeneity between clients, meaning that clients with significant differences in data distribution will get less benefits when collaborating. However, since clients can't transmit real data to calculating data heterogeneity in FL setting, this trade-off is difficult to handle. Although previous pFL works(Li et al. (2021), Collins et al. (2021), Zhang et al. (2021)) proposed different solutions from multiple perspectives including model distance, partial aggregation and knowledge transfer, these works are generally proposed based on the assumption of the static local data distribution which leads to the following issues when directly applied to FL scenarios of Flowing Data Heterogeneity under Restricted Storage: Firstly, existing pFL works typically estimate data heterogeneity between clients based on the information from local models, meaning that these pFL methods can not focus on the performance of the model on the inaccessible previous data, known as catastrophic forgetting(Kemker et al. (2018)). Thereby, the personalized global model obtained by the client may not necessarily meet its requirements(Sabah et al. (2023)). Secondly, client may meet the data of the same category that other clients have previously encountered during FL training, but the personalized global models obtained by the pFL methods under high data heterogeneity usually contains less global information, thereby slowing down the convergence rate of the model during FL training and reducing the generalization of the model on the data that may be encountered in the future (Zhu et al. (2021)). The issues above mean that existing pFL methods often perform poorly when directly applied to real-world scenarios.

Inspired by Continuous Learning (CL) based on generated replay(Zenke et al. (2017), Serrà et al. (2018)), we consider combining the pFL method with the data distribution replayed by the generator to achieve the goals above. Although there are already many Federated Continuous Learning (FCL) works(Qi et al. (2023), Ma et al. (2022), Zhang et al. (2023)) that combine FL with CL based on generated replay, the optimization objective of these FCL methods is to obtain a single optimal global model, meaning that directly applying these methods' replay generation scheme based on a single global generator to pFL methods with different optimization objective will result in two problems: Firstly, a single global generator is often difficult to replay the local data distribution of a specific client, making it difficult for pFL method to perform personalized aggregation based on the replayed distribution. Secondly, due to the low gradient similarity between clients under high data heterogeneity, the global generator requires more FL rounds to achieve convergence, there will be significant replay error in the early and middle stages of FL training(Li et al. (2023)). Since the global generator needs to mitigate catastrophic forgetting on its training by generated replay, the replay error will be further expanded, ultimately reducing the effectiveness of mitigating catastrophic forgetting and personalized aggregation. Therefore, we need to redesign the generated replay scheme to meet the requirements of pFL.

To address the challenges above, we propose our pFL framework: pFedGRP, to simultaneously achieve the goals of personalized aggregation, mitigating catastrophic forgetting and improving model generalization ability while protecting privacy. Due to the continuously arriving data over time under the FL setting of Flowing Data Heterogeneity under Restricted Storage which making it difficult to determine whether the model has converged, we focus on the comprehensive performance of the pFL method on the current and previous data distribution during each FL communication round rather than the final performance of the model obtained at the end of FL training. Then we attempt to solve the challenges above from both the data level and the model level. At the data level, in order to achieve the goals of reducing replay errors and controlling replay distribution, we design a local data distribution reconstruction scheme that effectively reduces the amount of replay data, then propose a category decoupled data generator architecture for the scheme to achieving the goals above and reducing training cost by partial updating. At the model level, we design a personalized aggregation scheme with learnable weights to flexibly trade-off the collaborative relationships between clients based on the low error local data distribution replayed by the local generator, then

we design a local knowledge transfer scheme to improve the generalization and convergence rate of the personalized global model. Our contributions can be summarized as follows:

1. We extend the optimization problem of pFL to the FL setting of Flow Data Heterogeneity under Restricted Storage where the FL methods focus on the comprehensive performance of the global model on all known local data distributions in each FL round during FL training.

2. We propose a local data distribution reconstruction scheme and a related category decoupled data generator architecture, then propose our pFedGRP framework with personalized aggregation and local knowledge transferring based on the replayed data distribution which is low error and controllable.

3. We conducted comparative experiments between our method and various FL, pFL, FCL methods on multiple benchmark datasets under various setting, and performed ablation experiments on our method. The experimental results validated the effectiveness of our pFL framework.

## 2    RELATED WORK

### 2.1    FEDERATED LEARNING AND PERSONALIZED FEDERATED LEARNING

Federated Learning (McMahan et al. (2017)) is a distributed machine learning paradigm that does not require the transmission of real data, the challenge faced in FL is how to aggregate the global model that performs well on all clients when the data distributions between clients are Non-IID. One approach to solving this challenge is to improve the performance of the global model by optimizing the knowledge transfer within the model space. Based on this approach, Li et al. (2020b) added a regularization term that penalizes the deviation between the local model parameters and the global model parameters during local training to improve convergence performance; Li et al. (2023) proposed fine-tuning the trainable aggregation weight on the validation set of the server to improve the generalization ability of the global model. Another approach to solving this challenge is to control the degree of collaboration between clients to improve the performance of the global model on each client, which is also known as personalized federated learning methods. Based on this idea, Marfoq et al. (2021) considered local data distribution as a weighted mixture of multiple underlying distributions, and calculates the weights of each sub model corresponding to each underlying distribution based on a EM algorithm on the client's local dataset; Ye et al. (2023) proposed constructing personalized client collaboration graphs based on cosine similarity of parameters between local models. However, the existing FL and pFL methods are designed based on the assumption of static local data distribution, meaning that they are difficult to achieve good performance when applied to the FL situation of Flowing Data Heterogeneity under Restricted Storage.

### 2.2    FEDERATED CONTINUE LEARNING BASED ON GENERATED REPLAY

The goal of Federated Continue Learning based on generative replay is to mitigate the negative impact of client environment changes on global model performance while protecting privacy, the challenges faced in FCL are mitigating catastrophic forgetting and transferring knowledge between tasks. One approach to addressing these challenges is to directly combine FL with CL by Weighted aggregating the local models obtained from local CL to achieve FCL. Based on this approach, Yoon et al. (2021) proposed decomposing the model into a weighted combination of global parameters for learning general knowledge and adaptive parameters related to the task to improve model performance; Liu et al. (2023) proposed a transformer based partial model component enhancement scheme to alleviate catastrophic forgetting Another approach to addressing these challenges is to obtain global knowledge through FL to assist in local CL. Using this approach, Babakniya et al. (2023) proposed a knowledge distillation scheme that trains a generator based on a global model on server to generate high-quality data for local replay of global features; Wuerkaixi et al. (2024) proposed to train local modes and local generator alternately based on the real data and the replay features of the global generator during the local training on the client side to extract data features, and send the local generator to the server for aggregation to update the global generator. Another way to addressing these challenges is to use model distillation to enable local models to acquire knowledge from other models. Based on this way, Dong et al. (2022) designed a distillation scheme based on class aware gradient compensation loss and class semantic relation distillation loss to ensure local cross task inter class relationship consistency; Qi et al. (2023) proposed a knowledge

distillation scheme based on the ACGAN model which uses generative replay for feature alignment and consistency enhancement during local training and global fine-tuning stages. Compared with the works above, the goal of our method is to customize personalized global models for each client rather than training a model that performs well globally, meaning that these works are basically orthogonal to our work.

# 3 PRELIMINARY

In this section, we first define the symbols to be used in our work, then elaborate on the optimization problem we need to solve. For the representation of the models, we use $C$ to represent the model used to solve practical problems (referred to as the Task Model), with its parameters denoted as $\theta_C$, and use $A$ to represent the model used to generate replay (referred to as the Auxiliary Model), with its parameters denoted as $\theta_A$. For the representation of the distribution and the data, we use $\mathcal{P} = (\mathcal{X}, \mathcal{Y})$ to represent the joint distribution $\mathcal{P}$ of the distributions $\mathcal{X}$ and $\mathcal{Y}$, use $\mathcal{P}_1 \& \mathcal{P}_2$ to represent the weighted mixture of two distributions $\mathcal{P}_1, \mathcal{P}_2$ based on the data volume of each distribution, use $\&_{i=1}^{n} \mathcal{P}_i$ to represent the weighted mixture of n distributions $\mathcal{P}_1, ..., \mathcal{P}_n$ based on the data volume of each distribution, and use $\mathcal{D}_1 \cup \mathcal{D}_2$ to represent the merging of two datasets $\mathcal{D}_1, \mathcal{D}_2$.

## 3.1 NOTATIONS AND PROBLEM FORMULATION

**Federated Learning and Personalized Federated Learning**: Assuming there are $n$ clients participating in FL, the set of clients is denoted as $\mathcal{C} = \{\mathcal{C}_1, \ldots, \mathcal{C}_n\}$. For each client $\mathcal{C}_i \in \mathcal{C}$, we use $\mathcal{P}_{\mathcal{C}_i} = (\mathcal{X}_{\mathcal{C}_i}, \mathcal{Y}_{\mathcal{C}_i})$ to represent its local data distribution, and use $C_i$ and $C_{*,i}$ to represent the local task model uploaded to the server and the global task model received from the server whose model parameters are denoted as $\theta_{C_i}$ and $\theta_{C_{*,i}}$. The Federated Learning methods aggregate the local task model parameters $\{\theta_{C_i}\}_{i=1}^{n}$ of each client to obtain a global task model $C_g$ whose parameter is denoted as $\theta_{C_g}$ that minimizes the expected value of task driven loss $\mathcal{L}(\cdot, \cdot)$ on the local data distributions $\{\mathcal{P}_{\mathcal{C}_1}, \ldots, \mathcal{P}_{\mathcal{C}_n}\}$ (i.e. $\theta_{C_{*,i}} = \theta_{C_g}$). The personalized Federated Learning methods aggregate a personalized global task model $C_{g,i}$ whose parameter is denoted as $\theta_{C_{g,i}}$ for each client $\mathcal{C}_i$ that minimizes the expected value of $\mathcal{L}(\cdot, \cdot)$ on $\mathcal{P}_{\mathcal{C}_i}$ (i.e. $\theta_{C_{*,i}} = \theta_{C_{g,i}}$). Therefore, the optimization objectives of FL and pFL can be expressed as the following $F_1$:

$$F_1 = \left\{ \min_{\theta_{C_{*,i}}} \mathop{E}_{(x,y) \sim \mathcal{P}_{\mathcal{C}_i}} \left[ \mathcal{L}(\theta_{C_{*,i}}, (x,y)) \right], \forall \mathcal{C}_i \in \mathcal{C} \right\} \tag{1}$$

However, most existing methods on FL and pFL typically assume that each local data distribution $\mathcal{P}_{\mathcal{C}_i}$ are static in all $T$ communication rounds of FL, that is, for any FL round $t, t' \in \{1, ..., T\}$, it satisfies $\mathcal{P}_{\mathcal{C}_i}^{t} = \mathcal{P}_{\mathcal{C}_i}^{t'}, \forall \mathcal{C}_i \in \mathcal{C}$. Therefore, these methods usually only focus on the performance of the global model on the data distribution of currently accessible data.

**Continual Learning and Federated Continual Learning**: The Continuous Learning setting in a centralized training environment consists of a sequence $\mathcal{T} = \{\mathcal{T}^1, \ldots, \mathcal{T}^T\}$ of $T$ tasks in time series. when executing the $t$-th task $\mathcal{T}^t \in \mathcal{T}$, the real-time data distribution is denoted as $\mathcal{P}^t = (\mathcal{X}^t, \mathcal{Y}^t)$, and the actual data distribution is a mixture of the real-time data distributions $\&_{t'=1}^{t} \mathcal{P}^{t'}$ of the previous $t$ tasks, and it will not be possible to access the real data of the previous $t-1$ tasks during task $\mathcal{T}^t$. The goal of CL at each moment $t$ is to obtain a task model $C^t$ that performs well in the current task and can maintain the performance on all previous tasks. Federated Continuous Learning typically refers to the FL where the client's local training process is in a CL setting, and the task switching on the client occurs at the beginning of each FL round. If the instant local data distribution of client $\mathcal{C}_i$ in the $t$-th FL round is defined as $\mathcal{P}_i^t$, the local data distribution $\mathcal{P}_{\mathcal{C}_i}^t$ of client $\mathcal{C}_i$ is a weighted mixture of the real-time local data distributions of the previous $t$ FL rounds (i.e. $\mathcal{P}_{\mathcal{C}_i}^t = \&_{t'=1}^{t} \mathcal{P}_i^{t'}$). Due to the fact that different clients $\mathcal{C}_i, \mathcal{C}_j$ typically work in different working environments, their instant local data distributions $\mathcal{P}_i^t, \mathcal{P}_j^t$ are usually different during the same FL round $t$. The goal of FCL is to aggregate a global task model $C_g^t$ based on the locally trained model parameters $\{\theta_{C_1^t}, \ldots, \theta_{C_n^t}\}$ of each client in each FL round $t$ which can minimize the expected value of $\mathcal{L}(\cdot, \cdot)$ on the local data distributions $\{\mathcal{P}_{\mathcal{C}_1}^t, \ldots, \mathcal{P}_{\mathcal{C}_n}^t\}$ of all clients. Using $\theta_{C_g^t}$ to represent the parameters of $C_g^t$ on $t$-th FL round, the optimization objective of FCL is represented as the following $F_2$:

$$F_2 = \left\{ \min_{\theta_{C_g^t}} \mathop{E}_{(x,y)\sim\mathcal{P}_{\mathcal{C}_i}^t} \left[ \mathcal{L}(\theta_{C_g^t}, (x,y)) \right], \forall \mathcal{C}_i \in \mathcal{C}, \forall t \in \{1,...,T\} \right\} \quad (2)$$

**Problem Formulation**: To simplify the modeling of Flowing Data Heterogeneity under Restricted Storage, we consider the case where the local distribution on the client switches with FL rounds which is similar to the definition of FCL. That is, the instant local data distribution $\mathcal{P}_i^t$ of each client $\mathcal{C}_i$ within any FL round $t \in \{1, \ldots, T\}$ is static. At this point, the optimization objective of the pFL method is extended to aggregate a personalized global model $C_{g,i}^t$ for each client $\mathcal{C}_i$ that minimizes the expectation of $\mathcal{L}(\cdot, \cdot)$ on its local data distribution $\mathcal{P}_{\mathcal{C}_i}^t$. Using $\theta_{C_{g,i}^t}$ to represent the parameters of $C_{g,i}^t$ on $t$-th FL round, the optimization objective of pFL can be extended as the following $F_3$:

$$F_3 = \left\{ \min_{\theta_{C_{g,i}^t}} \mathop{E}_{(x,y)\sim\mathcal{P}_{\mathcal{C}_i}^t} \left[ \mathcal{L}(\theta_{C_{g,i}^t}, (x,y)) \right], \forall \mathcal{C}_i \in \mathcal{C}, \forall t \in \{1,...,T\} \right\} \quad (3)$$

## 3.2 Optimization Problem

The challenges of solving the optimization objective $F_3$ lies in the following two points: Firstly, each client $\mathcal{C}_i$ needs to alleviate the catastrophic forgetting caused by the inability to access the real samples corresponding to $\{\mathcal{P}_i^1, \ldots, \mathcal{P}_i^{t-1}\}$ during local training in each FL round $t \in \{2, \ldots, T\}$. Secondly, the local data distribution $\mathcal{P}_{\mathcal{C}_i}^t$ on each client $\mathcal{C}_i$ may vary with the FL round $t$, meaning that a mechanism needs to be designed to estimate the distribution changes between clients to help the server perform personalized aggregation for each client $\mathcal{C}_i$.

To address the first challenge, inspired by the generation replay based CL methods, we configure an auxiliary model $A_i$ for each client $\mathcal{C}_i$ that can generate replay the history feature distributions. Specifically, use $\mathcal{X}_{A_i}$ to represent the replayed feature distribution of the auxiliary model $A_i$, before the local training of task $\mathcal{T}_i^t$ begins, the local replay distribution $(\mathcal{X}_{A_i^{t-1}}, \mathcal{Y}_{\mathcal{C}_i}^{t-1})$ composed of $\mathcal{X}_{A_i^{t-1}}$ which replayed by $A_i^{t-1}$ and the local label distribution $\mathcal{Y}_{\mathcal{C}_i}^{t-1} = \&_{t'=1}^{t-1}\mathcal{Y}_i^{t'}$ is close to the local data distribution $\mathcal{P}_{\mathcal{C}_i}^{t-1}$ at task $\mathcal{T}_i^{t-1}$. Therefore, client $\mathcal{C}_i$ can train the local task model $C_i^t$ on the data distribution $\{(\mathcal{X}_{A_i^{t-1}}, \mathcal{Y}_{\mathcal{C}_i}^{t-1}) \& \mathcal{P}_i^t\}$ to alleviate the catastrophic forgetting on task $\mathcal{T}_i^t$, then obtain the optimal local task model $C_i^{t,*}$ whose model parameters are denoted as $\theta_{C_i^{t,*}}$. Finally, client $\mathcal{C}_i$ updates the auxiliary model $A_i^{t-1}$ to $A_i^t$ to replay the approximation of the local feature distribution.

To address the second challenge, we propose using auxiliary model $A_i^t$ to replay the approximation of $\mathcal{P}_{\mathcal{C}_i}^t$ (i.e. $(\mathcal{X}_{A_i^t}, \mathcal{Y}_{\mathcal{C}_i}^t)$) on the server to aggregate a personalized global model for client $\mathcal{C}_i$. Without loss of generality, we concretize the collaborative relationship between client $\mathcal{C}_i$ and other $n-1$ clients through weight vector $\boldsymbol{W}_i^t = \{w_{i,1}^t, \ldots, w_{i,n}^t\}$, then the server optimizes the aggregated weights for client $\mathcal{C}_i$ by minimizing the task driven loss of the personalized global model parameter $\sum_{j=1}^n w_{i,j}^t \theta_{C_j^{t,*}}$ which aggregated from the optimal task model parameters $\{\theta_{C_1^{t,*}}, \ldots, \theta_{C_n^{t,*}}\}$ on $(\mathcal{X}_{A_i^t}, \mathcal{Y}_{\mathcal{C}_i}^t)$. Finally, server aggregates the personalized global task model $C_{g,i}^t$ for client $\mathcal{C}_i$ based on the optimal aggregation weight $\boldsymbol{W}_i^{t,*} = \{w_{i,1}^{t,*}, \ldots, w_{i,n}^{t,*}\}$ (i.e. $\theta_{C_{g,i}^t} = \sum_{j=1}^n w_{i,j}^{t,*} \theta_{C_j^{t,*}}$). Now The optimization problem $F_3$ can be transformed into the following optimization problem $F_4$ for solving:

$$F_4 = \left\{ \min_{\boldsymbol{W}_i^t} \mathop{E}_{(x,y)\sim\left\{(\mathcal{X}_{A_i^t}, \mathcal{Y}_{\mathcal{C}_i}^t)\right\}} \left[ \mathcal{L}\left( \sum_{j=1}^n w_{i,j}^t \theta_{C_j^{t,*}}, (x,y) \right) \right], \forall \mathcal{C}_i \in \mathcal{C}, \forall t \in \{1,...,T\} \right\} \quad (4)$$

$$where\ \theta_{C_i^{t,*}} \leftarrow \mathop{argmin}_{\theta_{C_i^t}} \mathop{E}_{(x,y)\sim\left\{(\mathcal{X}_{A_i^{t-1}}, \mathcal{Y}_{\mathcal{C}_i}^{t-1}) \& \mathcal{P}_i^t\right\}} \left[ \mathcal{L}(\theta_{C_i^t}, (x,y)) \right]; s.t. \sum_{j=1}^n w_{i,j}^t = 1$$

However, there are still two challenges in efficiently solving optimization problem $F_4$: Firstly, the auxiliary model usually cannot fully fit the actual feature distribution(Feng et al. (2021)). Especially, as the number of tasks increases, it may underfit the distribution which caused by insufficient model parameters(Bubeck & Sellke (2021)), ultimately affecting the effectiveness of local training and personalized aggregation(Wang et al. (2024), Domingos (2012)). Secondly, even if the auxiliary model has sufficient parameters to fit the local feature distribution, it still needs to alleviate its

catastrophic forgetting on training by generating replay, and the larger auxiliary model also require longer training time and more computing resources to fit new feature distributions. In the next chapter, we will elaborate on how to solve the optimization problem $F_4$ in the face of the two challenges above.

# 4 METHODOLOGY

## 4.1 PROBLEM DECOMPOSITION

To address the two challenges mentioned above, we design a local data distribution reconstruction scheme that can effectively reduce the amount of replay data and an auxiliary model architecture corresponding to this scheme to improve the generate replay capability of the auxiliary model while reducing additional training costs.

**Local Data Distribution Reconstruction Scheme**: In machine learning, the statistical heterogeneity of data is mostly reflected in categories(Collins et al. (2021)). Thus, the local data distribution $\mathcal{P}_i = (\mathcal{X}_i, \mathcal{Y}_i)$ on client $\mathcal{C}_i$ can be regarded as the result of weighted mixing of the feature distribution $\mathcal{X}_{i,c=y}$ ($c$ refers to category) corresponding to data labeled $y \sim \mathcal{Y}_i$ based on the likelihood of the occurrence of that type of data. Using $Y_i^{t'}$ to represent the vector composed of the number of real data of each class in task $\mathcal{T}_i^{t'}$ for client $\mathcal{C}_i$, When the distribution replayed by the auxiliary model is close to the real feature distribution, client $\mathcal{C}_i$ can mix the data generated by $A_i^{t-1}$ based on the vector $Y_{\mathcal{C}_i}^{t-1} = \sum_{t'=1}^{t-1} Y_i^{t'}$ composed of the number of each type of real data that appeared in the previous $t-1$ tasks with the real data of task $\mathcal{T}_i^t$ to achieve the effect of approximating the data distribution $\{(\mathcal{X}_{A_i^{t-1}}, \mathcal{Y}_{\mathcal{C}_i}^{t-1}) \& \mathcal{P}_i^t\}$ to the local data distribution $\mathcal{P}_{\mathcal{C}_i}^t = \&_{t'=1}^t \mathcal{P}_i^{t'}$. However, when $t$ is large, this simple and crude generation replay method may lead to problems such as a large amount of training data and a small proportion of real data which bring more feature distribution error will ultimately affect the local training effect of the task model $C_i^t$.

To address the challenge above, we propose a Local Data Distribution Reconstruction Scheme based on label quantity scaling: In task $\mathcal{T}_i^t$, client $\mathcal{C}_i$ calculates the vector $Y_{\mathcal{C}_i}^t = \sum_{t'=1}^t Y_i^{t'}$ composed of the number of each type of data that has appeared in total $t$ known tasks, then proportionally shrink $Y_{\mathcal{C}_i}^t$ to a quantity where only one type of real data exists which is equal to the number of that type of data in $Y_i^t$. Using $Y_{\mathcal{C}_i}^{t,'}$ to represent the scaled down result of $Y_{\mathcal{C}_i}^t$, the vector $Y_{i,A}^t$ composed of the number of supplements for each type of data is the difference between $Y_{\mathcal{C}_i}^{t,'}$ and $Y_i^t$(i.e. $Y_{i,A}^t = Y_{\mathcal{C}_i}^{t,'} - Y_i^t$). However, when the client faces situations where the distribution changes significantly due to encountering new categories of data in a new task, the local label scaling scheme above will be difficult to reduce the amount of generated data then introduces significant distribution error to the local training of task model. Considering that the goal of generating replays is to alleviate the catastrophic forgetting of the task model during local training rather than further improving the task model's performance, we limit the number of generated data for each type to no more than the quantity of the most abundant type of real data in $Y_i^t$. The flowchart of our local data distribution reconstruction scheme is shown in Figure 2.

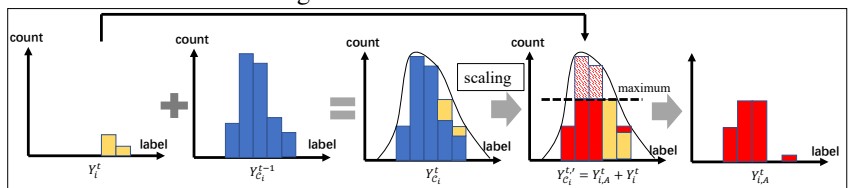

Figure 2: Flowchart of our local data distribution reconstruction scheme.

**Auxiliary Model Architecture**: As mentioned above, using a single auxiliary model will lead to insufficient model fitting ability(Bubeck & Sellke (2021)) and the need to alleviate the catastrophic forgetting effect of the auxiliary model itself. Given that there is currently no generative model that simultaneously possesses the characteristics of small model size, short training time and good generalization performance (Cao et al. (2023)), we consider that using a single auxiliary model to record the features of all types of data during local training on the client side is inefficient. Therefore, we propose decoupling the auxiliary model with respect to labels by establishing an auxiliary sub

model for each type of data encountered on the client. Specifically, in task $\mathcal{T}_i^t$, the auxiliary model $A_i^t$ on client $\mathcal{C}_i$ is a set of auxiliary sub model $A_{i,c}^t$ corresponding to each category $c \in \mathcal{Y}_{\mathcal{C}_i}^t$, denoted as $A_i^t = \left\{A_{i,c}^t\right\}_{c \in \mathcal{Y}_{\mathcal{C}_i}^t}$. Due to the small auxiliary sub model $A_{i,c}^t$ only needs to record the feature $\mathcal{X}_{i,y=c}$ of a single category $c$, $A_{i,c}^t$ hardly needs to consider alleviating catastrophic forgetting with generating replay when training with the real data of category $c$, and it can perform transfer learning on the previously trained auxiliary sub model of category $c$ on client $\mathcal{C}_i$ or other client $\mathcal{C}_j, j \neq i$ to effectively reduce the demand for computing resources and accelerating local training.

## 4.2 PFEDGRP

Based on the local data distribution reconstruction scheme and the label decoupled auxiliary model architecture mentioned above, we propose our pFL framework: pFedGRP, and take the $t \in \{1, \ldots, T\}$ FL round as an example to illustrate its process.

**Local Training**: Client $\mathcal{C}_i \in \mathcal{C}$ performs local training on task $\mathcal{T}_i^t$ in the $t$-th FL round, and has three models before training: The auxiliary model $A_i^{t-1,*}$ obtained by client $\mathcal{C}_i$ through local training in the previous FL round, the personalized global task model $C_{g,i}^{t-1}$ and the global task model $C_g^{t-1}$ aggregated by the server in the previous FL round. In FL round $t$, client $\mathcal{C}_i$ first calculates the vector $Y_{i,A}^t$ composed of the required number of generated data for each category based on the local data distribution reconstruction method above, then uses the auxiliary model $A_i^{t-1,*}$ to generate a replay dataset $\mathcal{D}_{A_i}^{t-1}$ based on $Y_{i,A}^t$, later mix it with the real data $\mathcal{D}_i^t \sim \mathcal{P}_i^t$ of task $\mathcal{T}_i^t$ to form the training dataset $\mathcal{D}_{A_i}^{t-1} \cup \mathcal{D}_i^t$ for the local task model. Considering that $C_{g,i}^{t-1}$ obtained by personalized aggregating is often difficult to contain a large amount of global information, we use the $C_g^{t-1}$ to initialize the local task model $C_i^t$ for local training while inheriting more global information, and align the outputs of $C_i^t$ and $C_{g,i}^{t-1}$ on the previously encountered categories of data to reduce feature drift and forgetting of previous tasks while preventing $C_i^t$ from distinguishing different categories of data based on differences between replay data and real data. Specifically, $C_i^t$ needs to minimize the difference between the output of $C_{g,i}^{t-1}$ and its output on the data of the previous category $c \in \mathcal{Y}_{\mathcal{C}_i}^{t-1}$. We define the alignment loss $\mathcal{L}_{align}$ based on mean square error ($MSE$) as follows:

$$\mathcal{L}_{align}\left(C_i^t, C_{g,i}^{t-1}, (x,y)\right) = \mathbf{1}_{y \in \mathcal{Y}_{\mathcal{C}_i}^{t-1}} MSE\left(C_i^t(x), C_{g,i}^{t-1}(x)\right) \quad (5)$$

Where $C(x)$ represents the output result of the model $C$ on data x, $MSE(\cdot, \cdot)$ represents the mean square error between two inputs, and $\mathbf{1}_*$ represents the indicative function with condition *. Finally, the local optimization objective is expressed as the following optimization objective $F_5$:

$$F_5 = \min_{\theta_{C_i^t}} \left( \sum_{(x,y) \in \left\{\mathcal{D}_{A_i}^{t-1} \cup \mathcal{D}_i^t\right\}} \left[ \mathcal{L}(\theta_{C_i^t}, (x,y)) + \lambda_{align} \cdot \mathcal{L}_{align}\left(C_i^t, C_{g,i}^{t-1}, (x,y)\right) \right] \right) \quad (6)$$

Where $\lambda_{align}$ controls the weight of alignment loss. Solving the optimization problem $F_5$ can obtain the parameters $\theta_{C_i^{t,*}}$ of the local optimal task model $C_i^{t,*}$. Then, the client $\mathcal{C}_i$ extracts the real data of each category $c \in \mathcal{Y}_i^t$ from $\mathcal{D}_i^t$ to train the corresponding auxiliary sub models while the auxiliary sub models of other categories directly use the previously trained results. Defining the real dataset corresponding to the category $c \in \mathcal{Y}_i^t$ is $\mathcal{D}_{i,y=c}^t$, the model parameter of the auxiliary sub model $A_{i,c}^t$ is $\theta_{A_{i,c}^t}$, and the training loss function is $\mathcal{L}_A$, then the update process of the auxiliary sub models is expressed as the following optimization objective $F_6$:

$$F_6 = \left\{ \min_{\theta_{A_{i,c}^{t-1,*}}} \left( \sum_{(x,y) \in \mathcal{D}_{i,y=c}^t} \left[ \mathcal{L}_A(\theta_{A_{i,c}^{t-1,*}}, x) \right] \right), \forall c \in \mathcal{Y}_{\mathcal{C}_i}^t \right\} \quad (7)$$

Solving the optimization problem $F_6$ can obtain the parameter set $\{\theta_{A_{i,c}^{t,*}}\}_{c \in \mathcal{Y}_i^t}$ of the optimal auxiliary sub model set $\{A_{i,c}^{t,*}\}_{c \in \mathcal{Y}_i^t}$. For other known categories $c' \notin \mathcal{Y}_i^t$, we directly use $A_{i,c'}^{t-1,*}$ obtained from the previous FL round as the optimal auxiliary sub model $A_{i,c'}^{t,*}$. For the case where encountering data of a new category $c''$, due to client $\mathcal{C}_i$ uninitialized the auxiliary sub model parameter $\theta_{A_{i,c''}^{t-1,*}}$, it will try to request the parameter cache $\theta_{A_{j,c''}^{t-1,*}}$ stored in the server uploaded by

other client $\mathcal{C}_j$ for transfer learning. If other clients have also not encountered data corresponding to category $c''$, a longer initialization training is performed on client $\mathcal{C}_i$ to obtain $\theta_{A_{i,c''}^{t-1,*}}$.

**Personalized Aggregation**: On the server side, the server receives the local task model parameters $\{\theta_{C_1^{t,*}}, \ldots, \theta_{C_n^{t,*}}\}$, local auxiliary sub model parameters $\left\{\{\theta_{A_{1,c}^{t,*}}\}_{c\in\mathcal{Y}_1^t}, \ldots, \{\theta_{A_{n,c}^{t,*}}\}_{c\in\mathcal{Y}_n^t}\right\}$, and local label distribution $\{\mathcal{Y}_{\mathcal{C}_1}^t, \ldots, \mathcal{Y}_{\mathcal{C}_n}^t\}$ uploaded by all $n$ clients in in the $t$-th FL round, and then solves the optimal personalized aggregation weight for each client $\mathcal{C}_i \in \mathcal{C}$. Without loss of generality, for client $\mathcal{C}_i$, the server first updates the auxiliary model cache corresponding to client $\mathcal{C}_i$ with auxiliary submodel parameters $\{\theta_{A_{i,c}^{t,*}}\}_{c\in\mathcal{Y}_i^t}$ to synchronize $A_i^{t,*}$ to the server, then samples the dataset $\mathcal{D}_{A_i}^t$ from the replay distribution $(\mathcal{X}_{A_i^{t,*}}, \mathcal{Y}_{\mathcal{C}_i}^t)$, finally minimizes the task driven loss $\mathcal{L}(\cdot, \cdot)$ of the aggregated model in $\mathcal{D}_{A_i}^t$, expressing as the following optimization objective $F_7$:

$$F_7 = \min_{\boldsymbol{W}_i^t} \sum_{(x,y)\in\mathcal{D}_{A_i}^t} \mathcal{L}\left(\sum_{j=1}^n w_{i,j}^t \theta_{C_j^{t,*}}, (x,y)\right), \ s.t. w_{i,j}^t \geq 0, \forall j; \sum_{j=1}^n w_{i,j}^t = 1 \quad (8)$$

Solving the optimization problem $F_7$ can obtain the optimal personalized aggregation weight $\boldsymbol{W}_i^{t,*}$, then server aggregates the personalized global task model $C_{g,i}^t$ for client $\mathcal{C}_i$ (i.e. $\theta_{C_{g,i}^t} = \sum_{j=1}^N W_{i,j}^{t,*}\theta_{C_j^{t,*}}$): Finally, the server uses local optimal models to average aggregate a global task model $C_g^t$ as the initialization model of the next round of local training for each client (i.e. $\theta_{C_g^t} = \frac{1}{n}\sum_{i=1}^n \theta_{C_i^{t,*}}$). The algorithm details and flowchart of pFedGRP can be found in Appendix C.1, and more discussion on Appendix F.

## 5 EXPERIMENT

### 5.1 EXPERIMENTAL PREPARATION

**Datasets**: We construct the FL setting of Flowing Data Heterogeneity under Restricted Storage based on existing datasets: For all datasets, we set the total number of clients to 10. For the MNIST, FashionMNIST and Cifar10 dataset with 10 categories, each client randomly divides these 10 categories into 5 tasks that each task consists of data from two categories and each category contains 200 real data. For the Cifar100 dataset with 100 categories and the EMNIST-ByClass dataset with 62 categories, each client randomly divides the categories into disjoint tasks by grouping them into two categories (i.e. 50 tasks for the CiFar100 dataset and 31 tasks for the EMNIST-ByClass dataset), with each category contains 200 real data. In our experiment, two adjacent tasks on the client switch after the server sends the aggregated model. Each training data in the dataset only appears in one FL round on each client, but the corresponding test data will be used in the testing of subsequent tasks. We provide detailed information on the dataset and training settings in Appendix A. For pFedGRP, We selected two classic generative replay models as auxiliary sub models based on the complexity of the dataset: the WGAN-GP(Cohen et al. (2017)) model with a network structure which is similar to DCGAN(Radford et al. (2016)) is chosen for MNIST series dataset, and the DDPM(Ho et al. (2020)) model sampled with DPM solver(Lu et al. (2022)) is chosen for Cifar series dataset.

**Baselines and Metrics**: We compare our pFedGRP with various FL, pFL and FCL baseline methods. For FL methods, we choose two classic methods: FedAVG(McMahan et al. (2017)), Fed-Prox(Li et al. (2020b)) and a FL concept drift method FedDrift(Jothimurugesan et al. (2023)); For pFL methods, we choose a classic FedEM(Marfoq et al. (2021)) and a newer pFedGraph(Ye et al. (2023)); For FCL methods, we choose four methods based on generate replay and model distillation: FedCIL(Qi et al. (2023)), TARGET(Zhang et al. (2023)), MFCL(Babakniya et al. (2023)), AF-FCL(Wuerkaixi et al. (2024)). We provide details of these methods in Appendix B. For evaluation metrics, we define Instant Average Accuracy (IAA) to measure the performance of each method in the current FL round, and calculate the Average Accuracy (AA) and Average Forgetting Measure (AFM) based on IAA to evaluate the overall effectiveness of the methods above. In short, the higher the average accuracy, the better the performance of the method. When the average accuracy of the two methods is close, the lower the average forgetting metric, the stronger the robustness of the method. We provide details of the metrics in Appendix C.2.

## 5.2 BASELINE EXPERIMENTS

We designed experiments based on the previous FL settings to compare pFedGRP with baseline FL methods in three scenarios. The first two scenarios are conducted on the MNIST, FashionMNIST, and CiFar10 datasets, the last scenario is conducted on the EMNIST-ByClass and Cifar100 datasets. Due to the FL setting of Flowing Data Heterogeneity under Restricted Storage where the client is unable to access the real data encountered in the previous task, each client can build up to 150 tasks on the MNIST and FashionMNIST datasets and up to 125 tasks on the Cifar10 dataset.

**FL with Tasks Gradually Changing**: In this setting, each client randomly selects two tasks from its five tasks (such as $\mathcal{T}_1$, $\mathcal{T}_2$) to form a task loop, that is, as the FL rounds increase, the client executes $\mathcal{T}_1, \mathcal{T}_2, \mathcal{T}_1, \mathcal{T}_2, \ldots$, and client randomly selects another task (such as $\mathcal{T}_3$) to replace one task in the task loop after executing 30 tasks (Cifar10 is 24 tasks). If task $\mathcal{T}_1$ is replaced, the task loop consists of $\mathcal{T}_2$ and $\mathcal{T}_3$. This setting corresponds to the common situation where the data distribution changes slowly in real-time. Our experimental results are reflected in Table 1 below:

Table 1: Baseline Experiment Results on FL with Tasks Gradually Changing

| FL methods | MNIST | | FashionMNIST | | Cifar10 | |
|---|---|---|---|---|---|---|
| | AA↑ | AFM↓ | AA↑ | AFM↓ | AA↑ | AFM↓ |
| FedAVG | 51.235 | 11.265 | 51.390 | 5.786 | 23.788 | 5.539 |
| FedProx | 57.702 | 8.900 | 56.618 | 4.969 | 23.472 | 4.391 |
| FedDrift | 22.071 | 8.641 | 21.008 | 6.999 | 18.268 | 6.893 |
| FedEM | 51.530 | 4.919 | 50.539 | 3.767 | 26.356 | 3.718 |
| pFedGraph | 54.597 | 10.026 | 54.49 | 4.164 | 22.638 | 4.090 |
| FedCIL | 76.692 | 0.522 | 74.167 | 0.573 | 31.222 | 0.839 |
| TARGET | 77.928 | 1.110 | 72.078 | 0.801 | 29.978 | 0.797 |
| MFCL | 76.167 | **0.306** | 70.852 | **0.387** | 29.135 | **0.280** |
| AF-FCL | 77.033 | 0.514 | 73.109 | 0.510 | 29.938 | 0.369 |
| pFedGRP(our) | **87.455** | 0.472 | **83.871** | 1.051 | **45.555** | 1.741 |

The reason why pFedGRP's overall performance can significantly lead other FL methods is that it can maintain the performance of the task model based on personalized aggregation before the FL model converges. After the FL model converges, its performance is similar to other FCL methods, and this performance is closely related to the replay effect of the auxiliary model. the IAA variation chart and corresponding experimental analysis are shown in Appendix E.1.

**FL with Tasks Circulating**: In this setting, each client grouped its five tasks into a task cycle, that is, as the FL rounds increased, the client executed $\mathcal{T}_1, \mathcal{T}_2, \mathcal{T}_3, \mathcal{T}_4, \mathcal{T}_5, \mathcal{T}_1, \ldots$. This setting corresponds to the situation where the data distribution changes extremely drastic which can better demonstrate the robustness of various FL methods. Our experimental results are reflected in Table 2 below:

Table 2: Baseline Experiment Results on FL with Tasks Circulating

| FL methods | MNIST | | FashionMNIST | | Cifar10 | |
|---|---|---|---|---|---|---|
| | AA↑ | AFM↓ | AA↑ | AFM↓ | AA↑ | AFM↓ |
| FedAVG | 67.780 | 7.961 | 54.681 | 4.333 | 21.061 | 3.129 |
| FedProx | 72.115 | 5.658 | 57.530 | 3.568 | 19.181 | 2.550 |
| FedDrift | 16.528 | 2.476 | 15.877 | 1.898 | 14.257 | 0.748 |
| FedEM | 70.729 | 5.990 | 56.390 | 3.596 | 19.083 | 3.180 |
| pFedGraph | 70.126 | 6.077 | 56.984 | 5.099 | 18.521 | 3.104 |
| FedCIL | 79.660 | 1.063 | 72.181 | 0.731 | 24.454 | 0.850 |
| TARGET | 77.255 | 0.975 | 70.355 | 1.676 | 18.644 | 0.423 |
| MFCL | 78.025 | **0.320** | 70.111 | **0.572** | 19.695 | **0.328** |
| AF-FCL | 78.740 | 0.902 | 70.890 | 0.667 | 21.984 | 0.561 |
| pFedGRP(our) | **89.437** | 1.277 | **81.845** | 0.845 | **40.595** | 0.790 |

The reason why the comprehensive performance of pFedGRP can significantly lead other FL methods is similar to the previous experiment, and the IAA variation chart and corresponding experimental analysis are shown in Appendix E.2.

**FL under High Data Heterogeneity**: We compared the performance of the above method under high data heterogeneity settings on the Cifar100 dataset and the EMNIST ByClass dataset. In this scenario, each client needs to complete a task loop consisting of all disjointed categories in the settings (Cifar100 includes 50 tasks and EMNIST-ByClass includes 31 tasks). At this point, all FL methods cannot converge, which better reflects the robustness of these methods. Our experimental results are shown in Table 3 below:

Table 3: Baseline Experiment Results on FL under High Data Heterogeneity

| FL methods | EMNIST-ByClass | | Cifar100 | |
|---|---|---|---|---|
| | AA↑ | AFM↓ | AA↑ | AFM↓ |
| FedAVG | 5.962 | 1.382 | 2.597 | 0.578 |
| FedProx | 6.233 | 1.418 | 2.573 | 0.563 |
| FedDrift | 3.204 | 0.603 | 2.065 | 0.399 |
| FedEM | 5.419 | 1.038 | 2.601 | 0.526 |
| pFedGraph | 7.364 | 2.718 | 3.331 | 1.330 |
| FedCIL | 5.754 | 0.971 | 1.867 | 0.327 |
| TARGET | 4.394 | 0.783 | 1.876 | 0.313 |
| MFCL | 4.917 | 0.658 | 1.530 | **0.213** |
| AF-FCL | 5.243 | **0.572** | 1.660 | 0.337 |
| pFedGRP(our) | **15.483** | 3.246 | **18.061** | 1.801 |

It can be seen that pFedGRP has stronger robustness in the case of not convergence, and the IAA variation chart and corresponding experimental analysis are shown in Appendix E.3.

**More Experiments**: We also conducted ablation experiments on pFedGRP framework and explored the performance changes of various FL methods under the setting of FL with Tasks Gradually Changing as the correlation between tasks gradually increased to verify the robustness of FL methods. Specific experimental details and results can be found in Appendix D.

## 6    CONCLUSION

In this paper, we attempt to solve the challenges of applying the pFL methods to the FL situation of Flow Data Heterogeneity under Restricted Storage. Based on the idea of low error generated replay, we propose a local data distribution reconstruction scheme that effectively reduces the number of generated data and a related class decoupled data generator architecture to achieve the goal of reducing data distribution replay errors and controlling replay data distribution. Then we propose our pFL framework: pFedGRP which composed of a personalized aggregation scheme based on replay distribution and a local knowledge transfer scheme improving the generalization of the task model. The effectiveness of pFedGRP has been validated in experiments with multiple datasets and settings.

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

# A   DATASETS

We use existing datasets to build the local dataset on the FL setting of Flowing Data Heterogeneity under Restricted Storage for each client. In our setting, the time interval between the server sends the global task model to the client is one FL round. The client executes one task in each FL round, and the data categories of tasks in the same FL round may be different between clients. The data categories of adjacent FL round tasks within the client are nonoverlapping in the baseline experiments. Each training data in the dataset only appears in one FL round on each client, but the corresponding test data will be used in the testing phase of the subsequent tasks. Therefore, we split each type of data on the training dataset into nonoverlapping parts, and proportionally split testing dataset as the test data for those corresponding parts. Each client includes the testing data corresponding to the new task's training data parts in its local test set when executing the new task. The schematic diagram of the partitioning of local training data and testing data are shown in Figure 3:

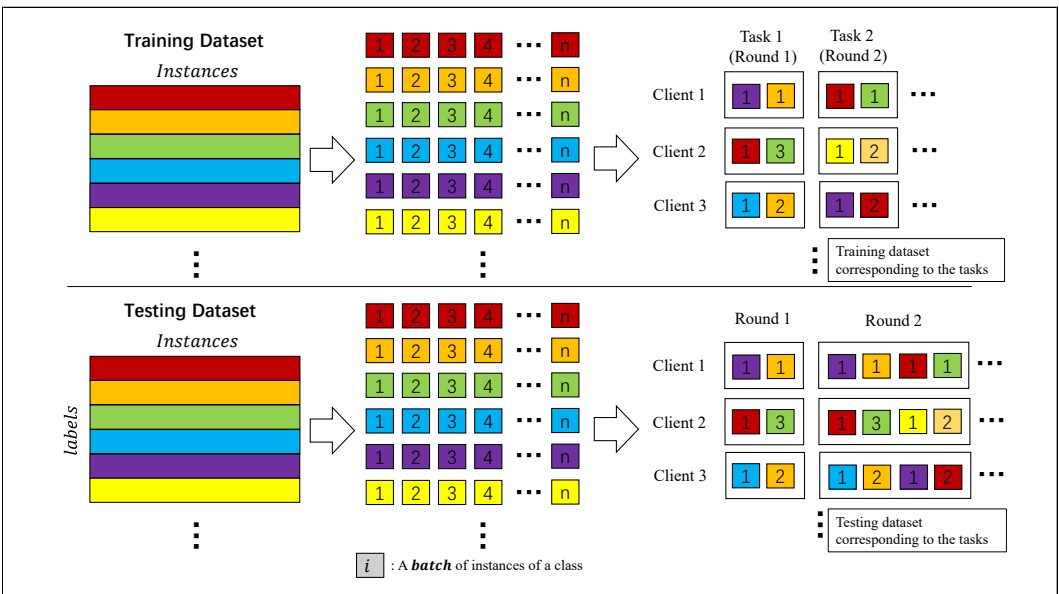

Figure 3: Schematic diagram of the partitioning of local training data and testing data.

The specific information of each dataset we used for the experiment is as follows:

**MNIST**. The MNIST dataset(LeCun et al. (1998)) is a 10 categories numerical classification dataset with 60000 training samples and 10000 test samples, and each sample is a single channel grayscale image with a size of 28x28 containing a number from 0 to 9. In our baseline experimental setup, the total number of clients is 10, each client contains 5 tasks, each task consists of 2 random and non repeating types of data with 200 data in each type.

**FashionMNIST**. The FashionMNIST dataset(Xiao et al. (2017)) is a clothing classification dataset consisting of 10 categories, each category with 6000 training samples and 1000 testing samples, and all samples are single channel grayscale images with a size of 28x28. Compared to the MNIST dataset, FashionMNIST dataset includes projections of objects from different perspectives which making it more challenging in terms of image quality and diversity. Our experimental setup on the FashionMNIST dataset is the same as that on the MNIST dataset.

**EMNIST-ByClass**. The EMNIST-ByClass dataset(Cohen et al. (2017)) is a dataset consisting of 62 imbalanced categories of handwritten characters and numbers with 814255 grayscale images of size 28x28. Compared with the MNIST dataset, EMNIST-ByClass dataset contains more categories, and its English character part includes uppercase and lowercase characters which increases the difficulty of classification. We strictly adhere to the definition of federated class incremental learning on this dataset: The total number of clients is 10, each client contains 31 tasks consisting of randomly non repeating two types of data with 200 training data and 100 testing data for each type.

**CIFAR10**. The CIFAR10 dataset(Krizhevsky & Hinton (2009)) is a real image classification dataset consisting of 10 categories of 32x32 color RGB images, each category containing 5000 training

images and 1000 test images. Compared with the MNIST series dataset, CIFAR-10 contains objects in the real world which have not only have a lot of noise but also different proportions and features, making data classification more difficult. Our experimental setup on the CIFAR10 dataset is the same as that on the MNIST dataset.

**CIFAR100**. The CIFAR100 dataset(Krizhevsky & Hinton (2009)) is a real image classification dataset consisting of 20 super categories, each super category has 5 categories and contains of 32x32 color RGB images. Each category contains 500 training images and 100 test images. Compared with the CIFAR10 dataset, the CIFAR100 dataset has a larger number of categories, and the images of each category within the same super category are more similar which increases the difficulty of classification. We strictly adhere to the definition of federated class incremental learning on this dataset: The total number of clients is 10, each client contains 50 tasks consisting of randomly non repeating two types of data with 200 training data and 100 testing data for each type.

## B  BASELINES DETAILS

We compare our personalized federated learning framework pFedGRP with following two FL methods, two pFL methods and four FCL methods. The FL methods and pFL methods do not have the ability to remember information related to historical tasks while the FCL methods can solve catastrophic forgetting and statistical heterogeneity problems. We additionally incorporated FL and pFL methods combined with our generative replay framework in the ablation experiment to validate the effectiveness of the personalized aggregation scheme of pFedGRP.

**FedAVG**: FedAVG(McMahan et al. (2017)) is a representative federated learning method, in which the server aggregates the task model parameters uploaded by each client based on the size of the client's local training set to obtain a global task model.

**FedProx**: FedProx(Li et al. (2020b)) is a classic federated learning method improved based on FedAVG, which adds a proximal term to the local training loss of each client to avoid the local task model deviating too much from the global task model. The aggregation strategy of the server on FedProx is consistent with FedAVG.

**FedDrift**: FedDrift(Jothimurugesan et al. (2023)) is a clustering federated learning method designed for distributed concept drift which divides the global data distribution into multiple domains. At the beginning of each FL round, clients calculate the local loss of each domain's global task model and compare the minimum loss with the last FL round's minimum loss to select an existing domain or create a new domain, then the server calculates the inter domain drifts based on the local loss and merges the domains with smaller drift by aggregating the corresponding models. Afterwards, clients perform local training on the task model of theirs corresponding domain and send local task model to server to aggregates global task model for each domain.

**FedEM**: FedEM(Marfoq et al. (2021)) is a classic personalized federated learning method that proposes the local data distribution is a weighted mixture of several underlying data distributions, and several task sub models are trained on each client to fit these underlying distributions. Then, the client performs EM steps on the local dataset based on several global task sub models aggregated by the server through FedAVG's strategy to calculate the personalized weights of each sub model.

**pFedGraph**: pFedGraph(Ye et al. (2023)) is a relatively new personalized federated learning method whose server uses the cosine difference degree between the local task model parameters to solve the inter client collaboration graph that can balance the relationship between individual utility and collaboration benefit to provide personalized aggregation of global task models for each client. During local training, the cosine similarity between the local task model and the personalized global task model from the previous round is constrained to prevent model bias.

**FedCIL**: FedCIL(Qi et al. (2023)) is a relatively new federated class incremental learning method which integrates the task model and auxiliary model into one ACGAN model. In the client local training phase, it adds a step of model distillation and label alignment on the data generated from the global ACGAN model and the previous local ACGAN model to alleviate catastrophic forgetting of the local ACGAN model. In the server aggregation phase, the local ACGAN models are first averaged aggregated to obtain the global ACGAN model, and then distill the global ACGAN model based on the generated data of each local ACGAN model.

**TARGET**: TARGET(Zhang et al. (2023)) is a relatively new federated class incremental learning method based on global feature replay. On the server side, it trains a global generator based on the BN layer features of the aggregated global task model and an untrained task model. On the client side, it alleviates the catastrophic forgetting of the task model based on the data replayed by the global generator.

**MFCL**: MFCL(Babakniya et al. (2023)) is a relatively new federated class incremental learning method based on global sample free replay and distillation. It proposed a scheme to training a global generator capable of generating high-quality data based on an aggregated global task model on the server side, and transfers the knowledge of the global task model to the local task model through distillation based on the generated data of the global generator during local training.

**AF-FCL**: AF-FCL(Wuerkaixi et al. (2024)) is a relatively new federated class incremental learning method based on local sample free replay which designs a local distillation mechanism based on partial feature forgetting. On the client side, it trains local task model and local auxiliary model alternately based on the real data and the data generated by global auxiliary model to achieve the goal of extracting data features for local task model while obtaining better replay effects for local auxiliary model. On the server side, average aggregation is used to aggregate global task model and global auxiliary model to obtain global information.

**FedAVG-replay**: The FedAVG algorithm that additionally uses the generate replay scheme of our pFedGRP during local training.

**pFedGraph-replay**: The pFedGraph algorithm that additionally uses the generate replay scheme and knowledge transfer scheme of our pFedGRP during local training.

## C  IMPLEMENTATION DETAILS

### C.1  ALGORITHM AND FLOWCHART OF PFEDGRP

The flowchart of pFedGRP's local training on client $\mathcal{C}_i \in \mathcal{C}$ on the $t$-th FL round is in Figure 4:

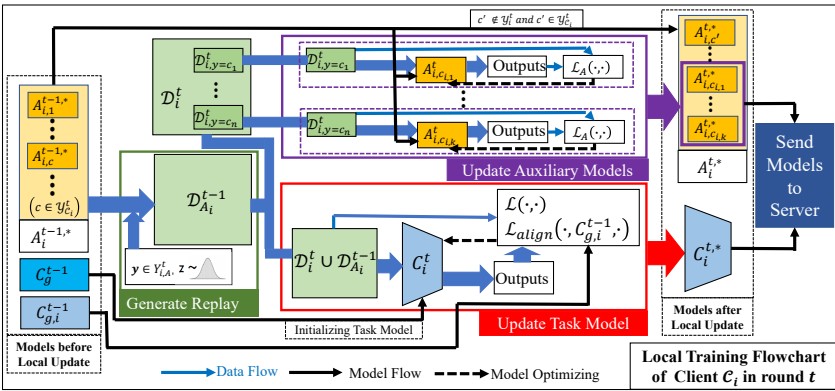

Figure 4: Local training flowchart of each client $\mathcal{C}_i$ under our pFedGRP framework.

The flowchart of pFedGRP's global aggregation on the server on the $t$-th FL round is in Figure 5:

The algorithm for pFedGRP is in Algorithm:

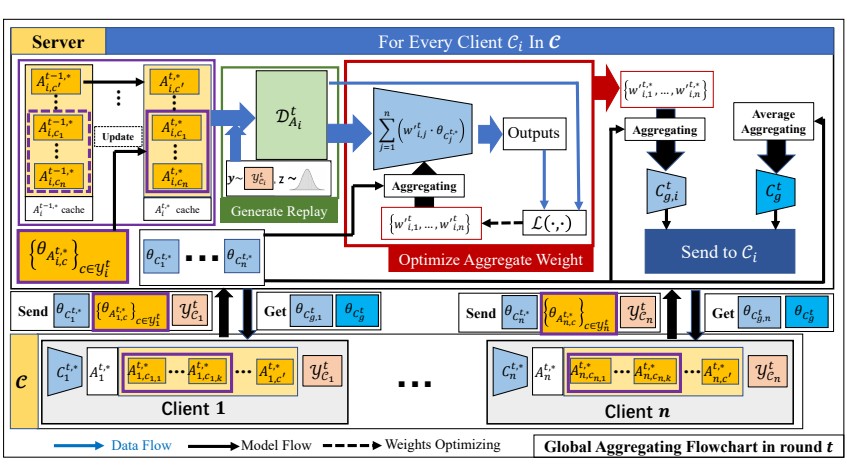

Figure 5: Personalized aggregation flowchart of server under our pFedGRP framework.

---

## Algorithm: pFedGRP

**Input**: Client set $\mathcal{C}$ with n clients; total round $T$; Task model param $\theta_C$, Auxiliary sub model param $\theta_A$ with randomly initialization;

**Output**: Model list $\left\{\theta_{C_{g,i}^t}\right\}_{i=1}^n$ of personalized global models corresponding to each client in each round $t \in \{1, \dots, T\}$

1    Server initializes $\theta_{C_{g,i}^0}, \theta_{C_g^0}$ for each client $\mathcal{C}_i \in \mathcal{C}$ with $\theta_C$.

2    **For each round** $t = 1, \dots, T$ **do:**

3      // Client local training

4      **For each client** $\mathcal{C}_i \in \mathcal{C}$ **in parallel do:**

5        Server send $\theta_{C_{g,i}^{t-1}}, \theta_{C_g^{t-1}}$ to $\mathcal{C}_i$, $\mathcal{C}_i$ initializes $\theta_{C_i^t}$ with $\theta_{C_g^{t-1}}$

6        **For each category** $c \in \mathcal{Y}_i^t$ **do:**

7          **If** $c$ previously appeared on other client $\mathcal{C}_j$ first appears on $\mathcal{C}_i$ **do:**

8            Server send $\theta_{A_{j,c}^{t-1,*}}$ from $A_j^{t,*}$ cache to $\mathcal{C}_i$ to initialize $\theta_{A_{i,c}^{t-1,*}}$

9          **End if**

10      $\mathcal{C}_i$ computes $\mathcal{Y}_{C_i}^t$ based on $\{Y_i^1, \dots, Y_i^t\}$

11      $\mathcal{C}_i$ computes $Y_{i,A}^t$ then constructs $\mathcal{D}_{A_i}^{t-1}$ based on feature replay distribution $\mathcal{X}_{A_i^{t-1,*}}$

12      $\mathcal{C}_i$ obtains $\theta_{C_i^{t,*}}$ by optimizing $F_5$ on $\left\{\mathcal{D}_{A_i}^{t-1} \cup \mathcal{D}_i^t\right\}$

13      $\mathcal{C}_i$ obtains $\left\{\theta_{A_{i,c}^{t,*}}\right\}_{c \in \mathcal{Y}_i^t}$ by optimizing $F_6$ on $\mathcal{D}_i^t$

14      $\mathcal{C}_i$ send $\theta_{C_i^{t,*}}, \left\{\theta_{A_{i,c}^{t,*}}\right\}_{c \in \mathcal{Y}_i^t}, \mathcal{Y}_{C_i}^t$ to server

15      **End For**

16      // Server aggregating

17      **For each client** $\mathcal{C}_i \in \mathcal{C}$ **do:**

18        Server updates $A_i^{t,*}$ cache with $\left\{\theta_{A_{i,c}^{t,*}}\right\}_{c \in \mathcal{Y}_i^t}$

19        Server constructs $\mathcal{D}_{A_i}^t$ based on replay distribution $\left\{\mathcal{X}_{A_i^{t,*}}, \mathcal{Y}_{C_i}^t\right\}$

20        Server optimizes $F_7$ on $\mathcal{D}_{A_i}^t$ then obtains $W_i'^{t,*}$

21        Server aggregates personalized global model param $\theta_{C_{g,i}^t} \leftarrow \sum_{j=1}^n \left(w_{i,j}'^{t,*} \cdot \theta_{C_j^{t,*}}\right)$

22      **End For**

23      Server aggregates global model param $\theta_{C_g^t} \leftarrow \frac{1}{n} \sum_{i=1}^n \theta_{C_i^{t,*}}$

24    **End For**

## C.2 Evaluation Metrics

We evaluate the performance of each method based on Instant Average Accuracy (IAA), Average Accuracy (AA) and Average Forgetting Measure (AFM). Assuming the client set is $\mathcal{C}$ and the total number of FL rounds is $T$, the definitions of the above metrics are as follows:

**Instant Average Accuracy**. After global aggregation in each FL round $t$, we evaluate the performance of the global task models on all test data corresponding to previous $t$ tasks on each client $\mathcal{C}_i \in \mathcal{C}$(i.e. accuracy, denoted as $a_i^t$), then calculate the IAA value of the $t$-th FL round based on the weighted average of the total number of training data encountered by each client $\mathcal{C}_i$ (denoted as $n_i^t$):

$$IAA^t = \frac{1}{\sum_{\mathcal{C}_i \in \mathcal{C}} n_i^t} \sum_{\mathcal{C}_i \in \mathcal{C}} n_i^t \cdot a_i^t \tag{9}$$

IAA can indicate the comprehensive performance of the global task model obtained in a certain FL round $t$ on all previous tasks.

**Average Accuracy**. This metric indicates the average performance of each method over the entire FL process based on the mean of the IAA values of all $T$ FL rounds:

$$AA = \frac{1}{T} \sum_{t=1}^{T} IAA^t \tag{10}$$

AA can reduce the evaluation error caused by changes in task difficulty to better evaluate the performance stability of different FL methods throughout the entire FL process.

**Average Forgetting Measure**. In continuous learning, the forgetting measure can be expressed as the degree to which the accuracy of the current task decreases compared to the previous task. We define the average forgetting measure as the average of the forgetting measure of the entire FL process:

$$AFM = \frac{1}{T-1} \sum_{t=2}^{T} max(0, \ IAA^{t-1} - IAA^t) \tag{11}$$

AFM can evaluate the degree of knowledge backward transfer, and the smaller the value, the better the memory stability of the FL method.

## C.3 Detailed Description of Experimental Setup

For the task model, we choose ResNet20(He et al. (2016)) as the task model for all FL methods except FedCIL. The local training rounds were uniformly set to 20, the optimizer was uniformly selected as SGD, the learning rate was set to 0.01, the momentum was set to 0.9, and the weight decay was set to 0.01. The ACGAN model of the FedCIL method adopts its default settings for each dataset with a local training round of 400.

For the auxiliary model, our method pFedGRP performs 1000 rounds of initialization training and 100 rounds of transfer learning on the MNIST series dataset corresponding to each category of WGAN-GP model on local training, and performs 6000 rounds of initialization training and 600 rounds of transfer learning on the Cifar series dataset corresponding to each category of DDPM model on local training. The training for auxiliary models of other FCL methods adopts the default settings corresponding to each dataset.

For the fine-tuning rounds during global aggregation, our method pFedGRP performs 20 rounds of personalized aggregation weight optimization for each client, the FedCIL method performs 100 rounds of model distillation on the global ACGAN model, and other FL methods do not have a fine-tuning stage for global aggregation.

# D Additional Experimental Results

## D.1 Ablation Experiments

Our method mainly consists of two modules: 1. Feature generation replay based on local data distribution reconstruction scheme and a category decoupling generator architecture corresponding to the

scheme. 2. Local training based on global task model and output alignment, and the personalized aggregation based on replay distribution. We conducted ablation experiments on each point in two settings constructed based on the MNIST dataset and FMNIST dataset in the baseline experiment.

For the first point, we referred to the generation replay schemes of other FCL methods which generate an equal amount of random data as the real data at each epoch of local training, so the categories of the data obtained by this type of generation replay scheme are random and uncontrollable. Since we set each task having two categories in the baseline experiment, we replaced the auxiliary model with a single WGAN-GP model that doubles the number of parameters (implemented by doubling the number of channels in the model), and the number of rounds for initialization training and transfer training are also set to 1000 and 100, respectively. The category of the generated data during local training is determined by the personalized global task model obtained in the previous FL round, and the category of the generated data during personalized aggregation is determined by the local optimal task model obtained in the current FL round. We denote this method as pFedGRP-AS1. Under this method, the client usually needs to generate more data during the local training process, and its local auxiliary model needs to replay the data based on the previous round's local auxiliary model during training to alleviate catastrophic forgetting. When encountering new categories of data, the client is usually unable to directly use other client's auxiliary models as pretrain model for transfer learning. The above means that it will greatly reduce the training efficiency of the auxiliary model and achieve poor generation replay ability in the same training epochs as pFedGRP.

For the second point, due to the fact that our training scheme consists of two parts: local training based on global task model and output alignment, and personalized aggregation based on replay distribution, we tested the performance separately when removing a certain part. For the first part, we remove the output alignment of the local training and separately initialize the local task model with the global task model obtained in the previous round and the personalized global task model to verify the effectiveness of our local training. These two methods are respectively referred to pFedGRP-ASG and pFedGRP-ASP. For the second part, we combine the global aggregation schemes of FedAVG and pFedGraph with our local training process to validate the effectiveness of our personalized aggregation method. These two methods are respectively referred to FedAVG-replay and pFedGraph-replay.

The experimental results of the five ablation methods mentioned above and our pFedGRP method are shown in Table 4 and Table 5. The IAA variation chart and corresponding experimental analysis are shown in Appendix E.4:

Table 4: Ablation Experiment Results on FL with Tasks Gradually Changing

| FL methods | MNIST | | FashionMNIST | |
|---|---|---|---|---|
| | AA↑ | AFM↓ | AA↑ | AFM↓ |
| pFedGRP-AS1 | 82.594 | 1.072 | 70.542 | 1.528 |
| pFedGRP-ASG | 68.445 | 5.285 | 81.192 | 1.589 |
| pFedGRP-ASP | 78.925 | 5.089 | 80.570 | 2.078 |
| FedAVG-replay | 83.326 | 1.569 | 78.135 | 1.264 |
| pFedGraph-replay | 83.153 | 1.427 | 80.472 | **0.622** |
| pFedGRP(our) | **87.455** | **0.472** | **83.871** | 1.051 |

Table 5: Ablation Experiment Results on FL with Tasks Circulating

| FL methods | MNIST | | FashionMNIST | |
|---|---|---|---|---|
| | AA↑ | AFM↓ | AA↑ | AFM↓ |
| pFedGRP-AS1 | 86.847 | **0.592** | 77.899 | 0.767 |
| pFedGRP-ASG | 81.928 | 3.202 | 79.545 | 1.062 |
| pFedGRP-ASP | 86.194 | 1.656 | 78.909 | 0.836 |
| FedAVG-replay | 87.021 | 2.488 | 80.158 | **0.685** |
| pFedGraph-replay | 89.211 | 1.419 | 80.296 | 0.809 |
| pFedGRP(our) | **89.437** | 1.277 | **81.845** | 0.845 |

## D.2 BASELINE EXPERIMENTS ON FL WITH DIFFERENT CORRELATIONS BETWEEN TASKS

We further investigated the robustness of our pFedGRP method and various baseline methods on the setting of the first baseline experiment(i.e. FL with Tasks Gradually Changing) on the MNIST, FashionMNIST, and Cifar10 datasets under different task correlations. Due to the fact that the number of duplicate categories between adjacent tasks of the same client in the setting above is 0, we increased this value to 2, 4 and 6 (i.e. each task has 4, 6 and 8 categories respectively) while the number of real data for each category remains at 200. Due to the limited amount of data in the real dataset, as the heterogeneity of data between and within clients decreases, the total number of rounds in FL and the total number of tasks for each client decreases to 70, 50 and 30, respectively (for Cifar10 is 60, 40 and 30). The results of pFedGRP and other baseline methods in the various experimental settings mentioned above are presented in Tables 6, Tables 7 and Tables 8:

Table 6: Baseline Experiment Results on MNIST and FL with Tasks Gradually Changing

| FL methods | The number of duplicate categories between adjacent tasks for the same client | | | | | | | |
| | 0 | | 2 | | 4 | | 6 | |
| | AA↑ | AFM↓ | AA↑ | AFM↓ | AA↑ | AFM↓ | AA↑ | AFM↓ |
|---|---|---|---|---|---|---|---|---|
| FedAVG | 51.235 | 11.265 | 88.023 | 1.147 | 90.605 | 0.507 | 91.431 | 0.063 |
| FedProx | 57.702 | 8.900 | 88.987 | 0.757 | 91.688 | 0.355 | 91.759 | 0.057 |
| FedDrirt | 22.071 | 8.641 | 24.429 | 6.872 | 56.304 | 2.475 | 87.615 | 1.265 |
| FedEM | 51.530 | 4.919 | 87.166 | 1.070 | 90.810 | 0.562 | 91.741 | **0.032** |
| pFedGraph | 54.597 | 10.026 | 85.458 | 1.441 | 89.844 | 0.520 | 88.411 | 0.128 |
| FedCIL | 76.692 | 0.522 | 89.975 | 0.244 | 92.147 | 0.163 | 92.341 | 0.154 |
| TARGET | 77.928 | 1.110 | 86.875 | 0.332 | 89.535 | 0.182 | 89.506 | 0.192 |
| MFCL | 76.167 | **0.306** | 87.325 | **0.191** | 89.639 | **0.068** | 89.119 | 0.131 |
| AF-FCL | 77.033 | 0.514 | 88.103 | 0.214 | 91.439 | 0.109 | 93.396 | 0.148 |
| pFedGRP | **87.455** | 0.472 | **90.168** | 0.285 | **92.778** | 0.169 | **94.570** | 0.172 |

Table 7: Baseline Experiment Results on FashionMNIST and FL with Tasks Gradually Changing

| FL methods | The number of duplicate categories between adjacent tasks for the same client | | | | | | | |
| | 0 | | 2 | | 4 | | 6 | |
| | AA↑ | AFM↓ | AA↑ | AFM↓ | AA↑ | AFM↓ | AA↑ | AFM↓ |
|---|---|---|---|---|---|---|---|---|
| FedAVG | 51.390 | 5.786 | 75.608 | 3.100 | 83.704 | 0.572 | 84.614 | 0.076 |
| FedProx | 56.618 | 4.969 | 78.278 | 2.400 | 85.375 | 0.382 | 85.184 | 0.062 |
| FedDrift | 21.008 | 6.999 | 29.385 | 5.968 | 47.938 | 3.265 | 82.203 | 1.036 |
| FedEM | 50.539 | 3.767 | 75.601 | 2.766 | 84.221 | 0.423 | 85.360 | 0.189 |
| pFedGraph | 54.49 | 4.164 | 74.183 | 3.702 | 81.984 | 0.614 | 81.434 | 0.286 |
| FedCIL | 74.167 | 0.573 | 83.245 | 0.341 | 87.354 | 0.241 | 84.587 | 0.103 |
| TARGET | 72.078 | 0.801 | 81.472 | 0.425 | 86.439 | 0.326 | 83.935 | 0.112 |
| MFCL | 70.852 | **0.387** | 82.410 | **0.120** | 86.612 | **0.119** | 84.476 | **0.052** |
| AF-FCL | 73.109 | 0.510 | 83.146 | 0.312 | 87.792 | 0.287 | 85.413 | 0.089 |
| pFedGRP | **83.871** | 1.051 | **86.472** | 0.740 | **88.685** | 0.518 | **86.925** | 0.653 |

Table 8: Baseline Experiment Results on Cifar10 and FL with Tasks Gradually Changing

| FL methods | The number of duplicate categories between adjacent tasks for the same client | | | | | | | |
| | 0 | | 2 | | 4 | | 6 | |
| | AA↑ | AFM↓ | AA↑ | AFM↓ | AA↑ | AFM↓ | AA↑ | AFM↓ |
|---|---|---|---|---|---|---|---|---|
| FedAVG | 23.788 | 5.539 | 50.969 | 3.538 | 58.045 | 1.376 | 63.298 | 0.655 |
| FedProx | 23.472 | 4.391 | 52.600 | 2.767 | **59.433** | 1.002 | 64.197 | 0.346 |
| FedDrift | 18.268 | 6.893 | 22.607 | 4.330 | 39.247 | 2.196 | 52.154 | 0.568 |
| FedEM | 26.356 | 3.718 | 52.266 | 2.940 | 57.630 | 1.451 | **64.958** | 0.448 |
| pFedGraph | 22.638 | 4.090 | 50.153 | 3.743 | 56.698 | 1.511 | 62.368 | 0.549 |
| FedCIL | 31.222 | 0.839 | 39.572 | 2.032 | 44.585 | 0.627 | 44.573 | 0.424 |
| TARGET | 29.978 | 0.797 | 42.351 | 1.324 | 45.372 | 0.394 | 48.421 | 0.323 |
| MFCL | 29.135 | **0.280** | 45.918 | **0.125** | 46.212 | **0.196** | 46.498 | **0.214** |
| AF-FCL | 29.938 | 0.369 | 44.926 | 0.892 | 47.235 | 0.423 | 49.631 | 0.354 |
| pFedGRP | **45.555** | 1.741 | **55.388** | 1.614 | 55.460 | 0.820 | 55.758 | 0.469 |

It can be seen from the tables above that the performance improvement of the three FL methods, two pFL methods and our pFedGRP framework is significant on the MNIST and FashionMNIST datasets with the decrease of data heterogeneity. However, due to the need to train auxiliary model for the FCL methods, the number of rounds required for convergence may not necessarily decrease which makes the performance improvement of the four FCL methods not significant. On dataset with complex data distribution such as Cifar10, the data distribution replayed by the auxiliary model often deviates significantly from the real data distribution, resulting in almost no performance improvement for the four FCL methods when data heterogeneity is low. Our pFedGRP method which obtains personalized global model based on replay data distributions with large deviations also performs worse than the FL method and pFL method, but its performance still leads the FCL methods due to the effective reduction of the errors of replayed data distribution introduced during local training.

# E IAA VARIATION CHARTS FOR EXPERIMENTS

## E.1 IAA VARIATION CHARTS FOR TASKS GRADUALLY CHANGING

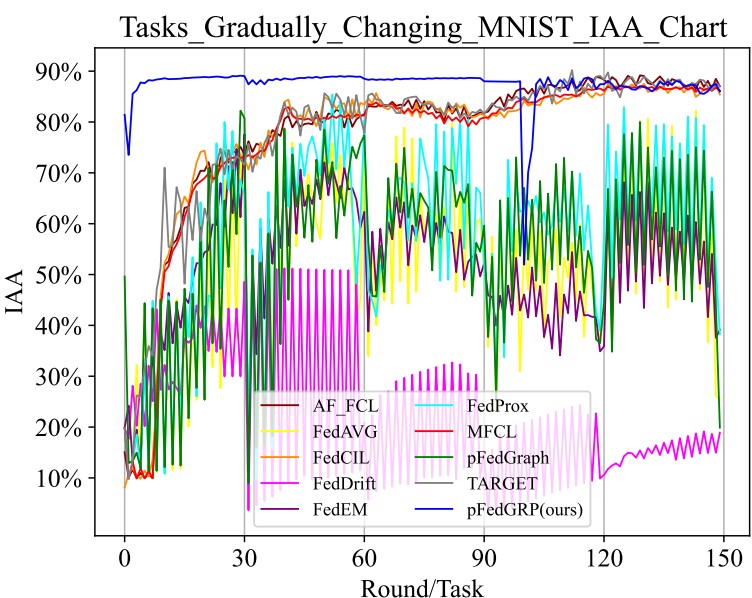

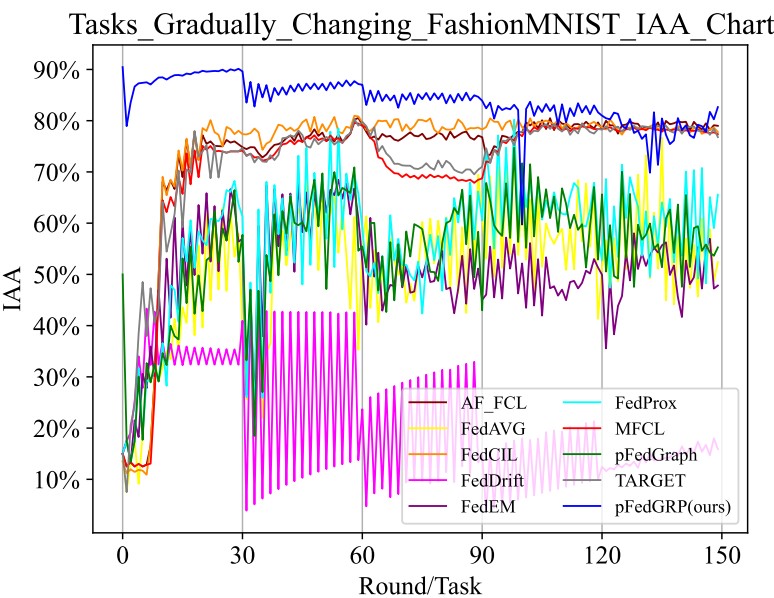

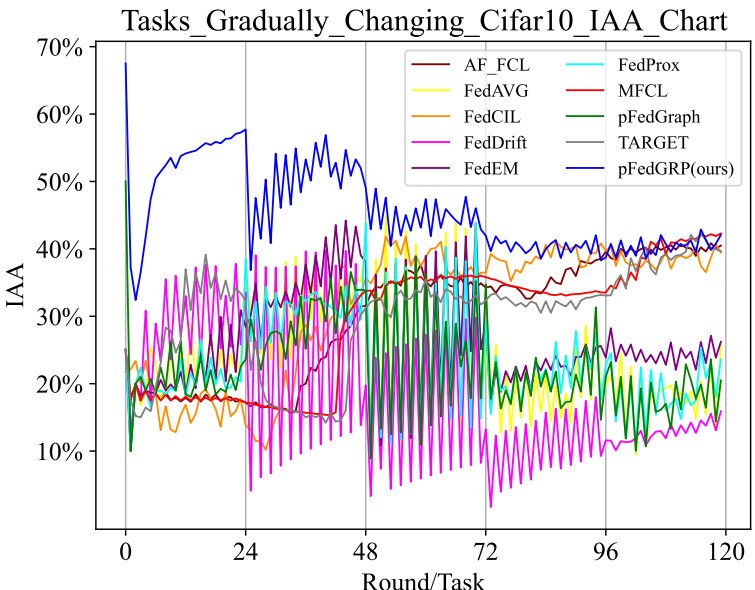

Under the FL setting of Tasks Gradually Changing, the gray vertical lines in the figure correspond to the FL rounds where the task set of each client's task loop changes. Firstly, overall, the pFedGRP method achieve good performance in the early stage and middle stage of FL training due to its ability to effectively estimate the data distribution of each client to aggregate personalized models for clients, and its performance in the later stage of training is not significantly different from other FCL methods, far superior to FL methods and pFL methods that do not have the ability to generate replay. Secondly, the pFedGRP method and the FCL methods in the baseline perform better on the MNIST dataset than the FashionMNIST dataset, and far better than the Cifar10 dataset which indicates that the performance of these methods is directly proportional to the quality of the data distribution replayed by the auxiliary model. Finally, due to the fact that the FCL methods in the baseline require training auxiliary model based on task model, the convergence time of these FCL methods is usually proportional to the data complexity of the dataset, resulting in poor performance in the early and middle stages of training. However, as a result, they often achieve stronger anti forgetting ability than pFedGRP after convergence.

## E.2 IAA VARIATION CHARTS FOR TASKS CIRCULATING

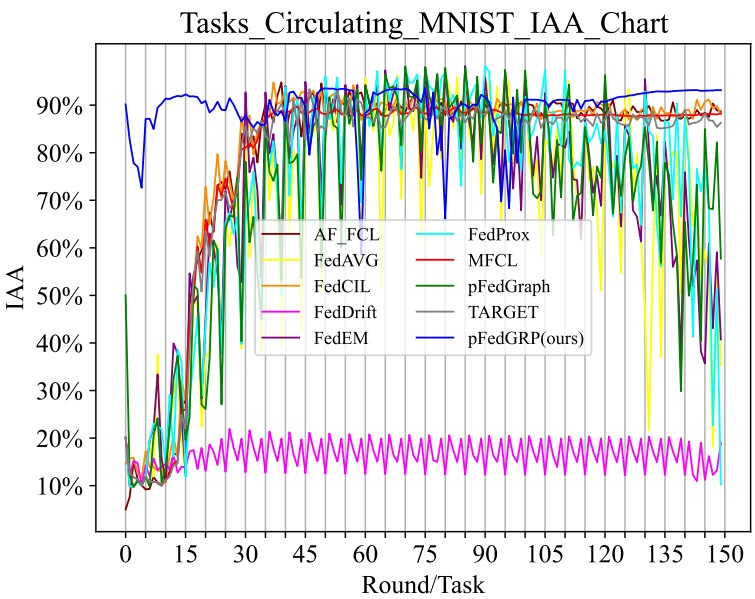

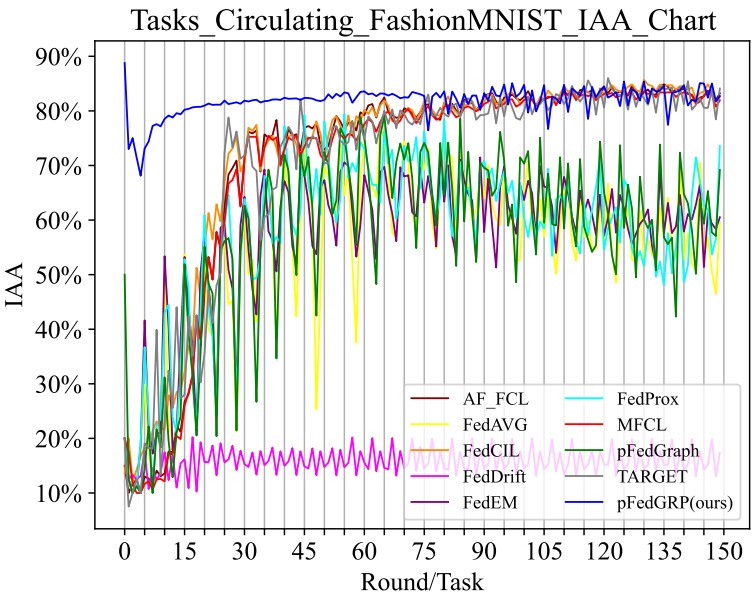

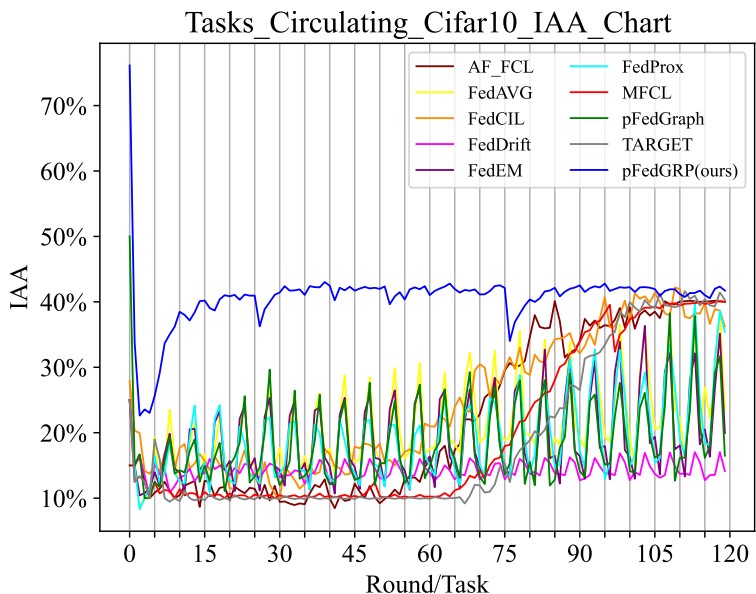

Under the FL setting of Tasks Circulation, the gray vertical line in the figure corresponds to the FL round at the beginning of each task cycle on each client (i.e. five rounds), which means that the distribution of data encountered by the client in every five rounds is similar to the data distribution of the entire FL process. The conclusion drawn from the experimental results under this setting is similar to that of the previous experiment.

### E.3 IAA VARIATION CHARTS FOR FL UNDER HIGH DATA HETEROGENEITY

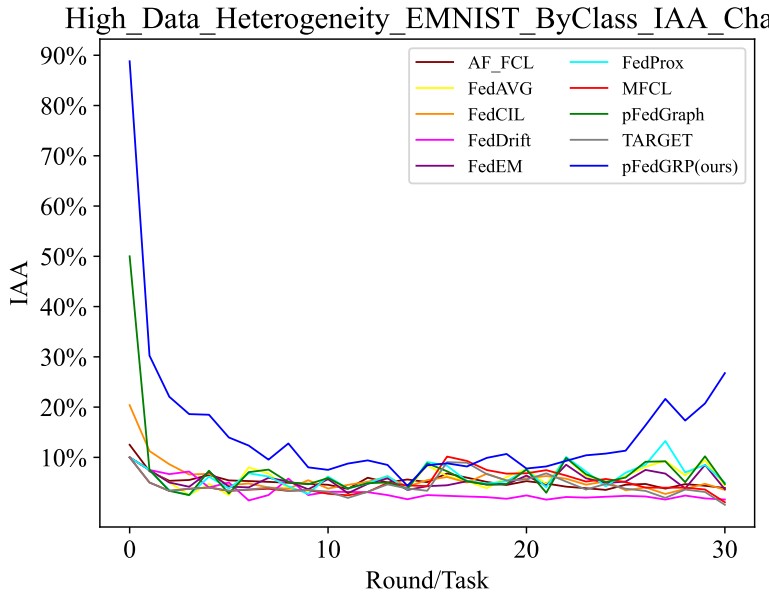

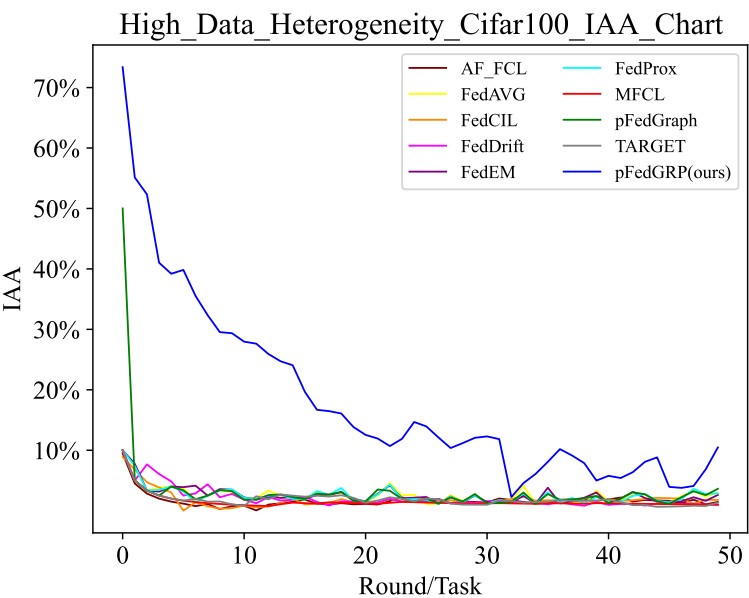

Under the FL setting of High Data Heterogeneity, each client encounter two categories of data that it has never encountered before in a new FL round until all categories in the dataset are traversed. This means that the FL setting in this experiment is similar to the one shot FL scenario which makes it impossible for all FL methods to converge, further testing the robustness of these FL methods. It can be seen that the pFedGRP method performs much better than other baseline methods when continuously encountering new categories.

### E.4 IAA VARIATION CHARTS FOR ABLATION STUDY

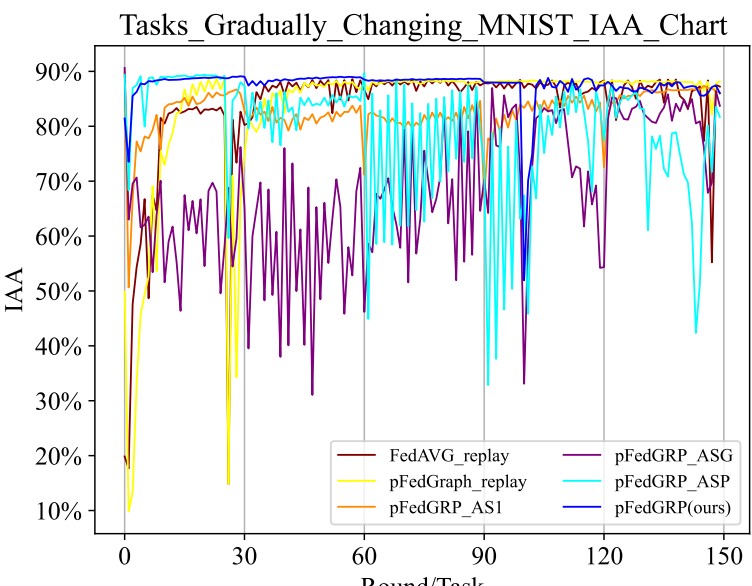

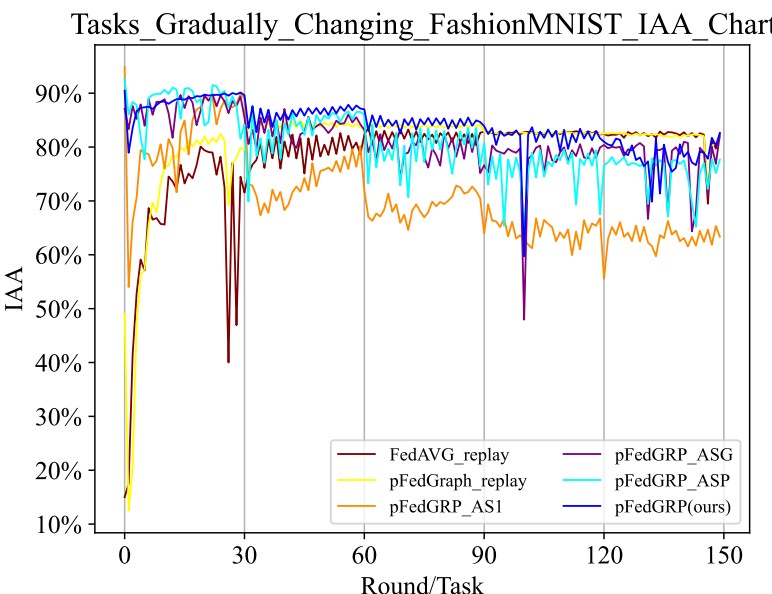

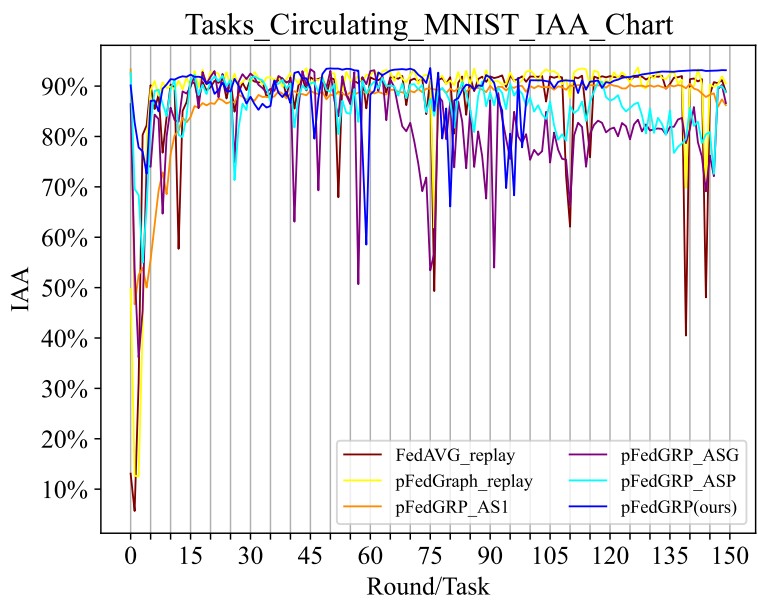

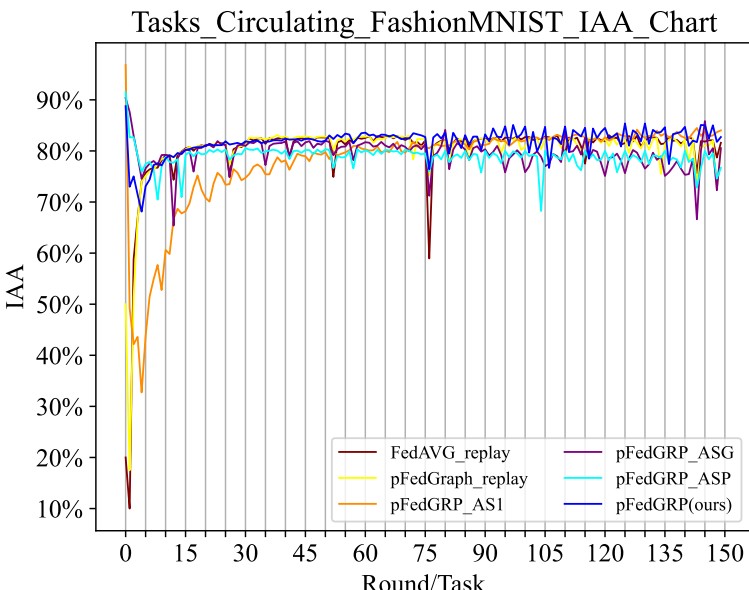

Firstly, it can be seen from the figure that the pFedGRP-AS1 method which use the generate replay scheme of other FCL methods achieved the worst results, indicating that the pFedGRP framework can achieve better results with less training consumption. Secondly, without using the local knowledge transfer scheme of the pFedGRP framework, the pFedGRP-ASG method which uses the global task model for local training performs worse than pFedGRP throughout the entire FL process, and the pFedGRP-ASP method which uses the personalized global task model for local training performs well in the early stages of FL training with fewer local data categories but worse than pFedGRP in the later stages of FL training with more local data categories, reflecting the effectiveness of the local knowledge transfer scheme of the pFedGRP framework. Without using the personalized aggregation scheme of the pFedGRP framework, FedAVG-replay and pFedGraph-replay perform worse than pFedGRP in the early stage of FL training but perform similarly to pFedGRP in the later stage of FL training after model convergence.

# F DISCUSSIONS

## F.1 ROBUSTNESS TO CHANGEABLE HETEROGENEITY LEVELS

The pFedGRP framework we proposed has strong robustness in the federated learning process, manifested in the following three aspects:

(1) Solving the optimal personalized aggregation weight based on low error replay distribution on the server can reduce the weight of task models for clients with large data distribution differences and improve the weight of task models for other clients with small data distribution differences. This enables the personalized global task model to enhance its generalization ability while ensuring model performance, and has natural robustness against model poisoning attacks.

(2) When the data distribution of the client undergoes significant changes in two adjacent FL rounds, the changes in its data distribution can be intuitively reflected in the distribution replayed by the auxiliary model, thereby causing the changing of the personalized aggregation weight to adapt to the changes in local data distribution.

(3) Even if some clients disconnect during the FL training process, due to the server-side storing the task models uploaded by the clients in the previous round of aggregation, the remaining clients can still perform personalized aggregation normally. Furthermore, if clients are allowed to use the latest historical task model caches of other clients on the server for personalized aggregation, our framework can be easily transformed into a asynchronous form.

## F.2 REDUCTION ON EXTRA TRAINING COST

The pFedGRP framework we proposed can reduce additional training burden while ensuring model performance, specifically manifested in the following three aspects:

(1) The auxiliary model on each client is essentially a collection of smaller sub models that record features of specific categories. These sub models only perform a small amount of transfer learning on the real data of the corresponding category in each round of local training to fit the features of the latest real data of that category. If there is no real data of that category, no training will be conducted, effectively reducing the additional training load.

(2) Due to the fact that it takes a long time for the client to train the auxiliary sub model of the category from scratch on the real data corresponding to the new category that other clients have already encountered, we send the auxiliary sub model cached on the server for this category to the client and conduct a small amount of transfer learning to effectively accelerating the local training speed of the client.

(3) The local data distribution reconstruction scheme we proposed can reduce the total number of local training data for the local task model on the client side while increasing the proportion of real data in local training data, which can speed up local training while reducing the error of the data distribution replayed by the local auxiliary model. Specifically, in common situations where similar categories of data are encountered repeatedly, it is possible to achieve the effect of making the reconstructed local data distribution approximate the local true data distribution. If the label distribution between tasks is very close, there is almost no need to generate data through auxiliary models to replay the data distribution.

## F.3 POTENTIAL OF HANDLING DIFFERENT TASKS

The pFedGRP framework we proposed does not make assumptions about the target of the task and does not limit the type of the task model and the auxiliary sub model. This means that our framework can choose different models according to different task requirements, thus having the potential to handle different tasks. However, some existing FL, pFL and FCL methods are specifically designed for specific types of tasks.

