# OpenReview forum: "Personalized Federated Learning on Flowing Data Heterogeneity under Restricted Storage"
_ICLR.cc/2025/Conference — ICLR 2025 Conference Withdrawn Submission_

### Official Review · Reviewer_Zys3 · 2024-10-18

**Soundness:** 3
**Presentation:** 3
**Contribution:** 2
**Rating:** 6
**Confidence:** 2

**Summary:**

This paper proposes pFedGRP to tackle data heteogeneity in federated learning (FL). The authors propose a local data distribution reconstruction scheme and a related category decoupled data generator architecture, then propose pFedGRP framework with personalized aggregation and local knowledge transferring based on the replayed data distribution which is low error and controllable. Extensive experiments validate the contribution.

**Strengths:**

The problem studied is interesting. The proposed algorithm is novel.

The authors provide solid experimental studies.

**Weaknesses:**

In terms of data heterogeneity in federated learning, the current version does not consider a popular benchmark setting [1]. For example, the Dirichlet-based label distributions (Dir(0.5), Dir(0.1), etc), and quantity-based label distributions (C=1, 2, 3, etc) can be further experimented on. The authors are suggested to discuss and experiment the proposed algorithm on the comprehensive non-IID settings. This would make the conclusion more convincing if the algorithm can achieve SOTA performance on various non-IID settings.

[1] Li, Qinbin, et al. "Federated learning on non-iid data silos: An experimental study." 2022 IEEE 38th international conference on data engineering (ICDE). IEEE, 2022.

**Questions:**

See weaknesses. I think the current version is on the borderline and will adjust based on the author's response.

---

> ### Author Response · Authors · 2024-11-20
>
> Thank you very much for taking the time to read our work and providing many suggestions! Below are our responses to the various weaknesses and questions you have raised:
>
> For weakness 1, we elaborate the correlation between our experimental setup and non-iid data setup in the following four points: 1. In our baseline experimental setup, the local data distributions of different clients are usually different and are only small parts of the global data distribution in each FL round (see lines 407-420 and Appendix A), which can be considered as an extreme non-iid data setting. 2. The local data distribution of each client is changed with FL rounds until the entire global data distribution is traversed, meaning that the non-iid data situation between clients are constantly changing (see lines 407-420 and Appendix A). 3. Under our setup, the client cannot access the real data encountered in all previous FL rounds during the local training process of each FL round, but the local test data contains the distribution of all previously encountered data (see lines 407-420 and Appendix A), which poses a much higher challenge than conventional static non-iid data settings. 4. We provide a comparative experiment in Appendix D.2 to increase the number of overlap categories between adjacent tasks from 0 to 6. Increasing the number of overlap categories not only reduces the degree of variation in local data distribution but also reduces the non-iid data settings between different clients.

---

> > ### Comment · Reviewer_Zys3 · 2024-11-22
> > **Ack of rebuttal**
> >
> > Thank for author's response. Given the other reviews, I keep my rating.

---

### Official Review · Reviewer_RaS3 · 2024-10-20

**Soundness:** 1
**Presentation:** 1
**Contribution:** 2
**Rating:** 1
**Confidence:** 4

**Summary:**

pFedGRP targets the challenge of data distribution change with respect to time for each client in a federated learning setting, where clients do not have sufficient space to store all the data samples. Combining a personalization strategy with continual learning, the authors propose (a) a scheme to reconstruct local data, (b) a local training algorithm based on the generated data, and (c) a global aggregation scheme to merge personalized models to achieve better generalization across clients.

**Strengths:**

The challenge of data distributions of each client changing with time (or federated rounds) is relevant and practical, this work is targeting a good problem.

**Weaknesses:**

1. The readability is low. Excluding the language-related mistakes, the technical content is often too hard to parse. Many symbols have 4-5 sub/superscripts (e.g., $\theta_{A^{t-1, *}_{j, c^{''}}}$), and it gets pretty hard to keep up with all those symbols. Instead of just having long paragraphs laying out the proposed algorithm, an actual algorithm (like in Appendix C) in the main content would have massively improved the readability.

2. Lack of having explored more relevant literature, and hence the lack of more relevant baselines. What the authors title "Flowing data heterogeneity" already is more popularly called "data drift", "concept drift" [1, 2], or "concept shift" [3] in literature. The authors claim that no personalization methods have looked into this challenge of "flowing data heterogeneity" (referring to line 67: "these works are generally proposed based on the assumption of the static local data distribution..."), that's because personalization methods target the challenge of "data heterogeneity" (different data distributions across the participating clients). But there are works [1, 2] (more can be found in the related work section of these two works) that do target the challenge of "changing data distribution with respect to time for all the client separately". Although I agree that if all clients change their local distributions, there would be "data heterogeneity across clients" as well, the authors have not compared against the baselines which can be closer to their work than the baselines only related to generating local personalized models.

[1] Federated Learning under Distributed Concept Drift (AIStats 2023)

[2] Flash: Concept Drift Adaptation in Federated Learning (ICML 2023)

[3] Advances and Open Problems in Federated Learning (FTML 2021)

3. The need of a "personalized global model" is unclear. Lines 164-165 say "the goal of our method is to customize personalized models for each client rather than training a model that performs well globally...", then why are each client's personalized models getting sent to the server? But then lines 95-96 state "the goals of personalized aggregation, mitigating catastrophic forgetting and improving model generalization ability,".

4. The claims about "privacy-preservation" (line 96) seem unfounded. If the auxiliary task models of each client are getting sent to the server, what if an adversary gets access to the auxiliary models and can generate similar data to a specific client's data? Line 933 (Algorithm line 7 in Appendix C) shows that each client's class auxiliary model can be transmitted to other clients as well.

5. The whole of the "Local Data Distribution Reconstruction Scheme" seems like a solution for a general continual learning (CL) method aiming to address catastrophic forgetting. It is unclear how that section is related to FL, or what specific challenges FL brings which can hinder using this type of solution.

6. The method is not scalable. These are the following models sent to each client each round: (a) A set of auxiliary models, (b) a personalized model specific to a client, and (c) an aggregated global model. Moreover, there are no communication (in terms of transmitted parameter count and size in MBs/GBs) and computation costs (a rough estimation of FLOPs) given in the manuscript, which seems essential to learn whether the performance benefits against the baselines are practical or not.

**Questions:**

1. In "Introduction", line 32 states "... to reduce \textbf{communication bandwidth} and \textbf{real-time requirements}". Can the authors clarify how clients uploading model weights to the server "reduces communication bandwidth"? And what are those "real-time requirements"? In practical deployments, one round can last from a few hours to a few days, so I am unsure about the "real-time" claim.

2. What is the Y-axis in Figure 1?

3. What does this sentence mean (lines 55-56): "FL methods should provide personalized global models for each client when \textbf{data heterogeneity is unknown}"?

4. Is Eq 1 the objective of both FL and pFL methods? Is it one client's objective or the server-side global model's training objective?

5. Can the proposed approach "remember" multiple past distributions? Or only the most recent previous distribution?

6. Related to the previous question (Q5), lines 243-244 state "client can train the local task model on the data distribution $({ \mathcal{X}^{t-1}}, \mathcal{Y}^{t-1}) + \mathcal{P}^t $" (excluding some other notations for brevity). But what if the past and the current distributions conflicting? (As an extreme case, what if the past distributions have all the dog photos tagged as "dog", but the current distributions have all the dog photos tagged as "cat"?).

7. Are there no other CL works that have used something similar to the "Local Data Distribution Reconstruction Scheme"?

8. Do all clients have to participate in each round?

9. What is the rationale behind the grouping strategy (lines 411-412) for larger-class-count datasets like CIFAR100 and EMNIST62?

10. What are the practical examples of "FL with Tasks Gradually Changing"? In other words, are there any examples where a client's data oscillates between two distributions, and then one distribution changes into a third one?

---

> ### Author Response · Authors · 2024-11-20
>
> Thank you very much for taking a lot of time reading our work carefully and providing many suggestions and questions! Below are our responses to the various weaknesses and questions you have raised:
>
> For Weakness 1, We apologize for the reading difficulties caused to you due to our limited expression ability.
>
> For weakness 2, our work is focusing on "Flowing Data Heterogeneity under Restricted Storage" rather than "Flowing Data Heterogeneity", the difference between the former and the latter is that the client in each FL round cannot access the training data encountered in all previous FL rounds, but needs to test the performance of the task model on the previously encountered test data. We think that the research field closer to this setting is Federated Continuous Learning rather than Federated Concept Drift. The baseline [1] provided by the reviewer proposed a domain clustering approach to address the problem of federated concept drift, we incorporate it into the experimental section (see Chapter 5.2, Appendix D.2, Appendix E). However, in our experimental setup, each client’s local training data distribution in each FL round is only a subset of the local testing data distribution, and the data categories corresponding to different clients' tasks in each FL round are usually different, this method creates too many domains then resulting in poor performance. Taking the dataset with 10 categories in the baseline experiment as an example (lines 406-419), each client randomly divides 10 categories into 10/2=5 tasks with non overlapping categories before FL training starts, meaning that there are $C_{10}^2 C_8^2 C_6^2 C_4^2 C_2^2/A_5^5=945$ different task combinations, making it difficult for different clients to have the same data category corresponding to tasks in the same FL round. As a result, method [1] generates many domains that are difficult to merge during FL training, clients switch their domain and perform local training during task switching, resulting in catastrophic forgetting and reduced performance on previous domains(tasks). Ultimately, this is reflected in an average accuracy rate close to 20\% ($2/10$). For baseline [2], as the GitHub link provided has expired, we did not include this method as a baseline method.
>
> For weakness 3, our statement here is incorrect. We have changed 'customized personalized models' to 'customized personalized global models'. Secondly, the local optimal task model sent by the client to the server is obtained by transferring the knowledge of the personalized global task model onto the global task model (see lines 345-362). For personalized global task model, it is aggregated by minimizing the loss on the client’s local replay data distribution (see lines 386-395). Under high data heterogeneity settings, it usually only contains global information of known categories of data on the client side, which makes it performs well on the client side but has poor generalization ability for unknown category data. For the global task model, it is composed of the average aggregation of local models (see lines 397-399), so it contains global information of known categories of data for all clients. However, under high data heterogeneity settings, it faces difficulties in convergence and performs poorly on clients. We combine the information from these two models to enhance the generalization ability of the global personalized model on the client when encountering new categories of data in the future, as other clients may have already encountered data of that category and passed this information to the client through the global task model.

---

> > ### Author Response · Authors · 2024-11-20
> >
> > For weakness 4, the privacy protection emphasized by federated learning mainly refers to the inability to expose the original local data of each client to other devices. Existing research has shown that the auxiliary model is almost impossible to fully record the content of real data (see line 226), always resulting in a replay error. It can be considered that the replayed data is the original data with privacy protection measures like differential privacy. This is also the reason why some existing FCL methods (such as FedCIL and AF-FCL in the baseline, see Appendix B) choose to upload auxiliary models to the server. In our approach, the goal of sending auxiliary sub models to other clients as an initialization model for transfer learning is to reduce the training cost of the auxiliary model. Assuming the total number of data categories is $c$, the total number of clients is $n$, if there are no overlapping categories between clients in the first FL round, the initialization training time for sending auxiliary models to other clients is $c$, and for not sending auxiliary models to other clients is $c \times n$, significantly reducing additional training costs. In the worst-case situation, the data categories between clients are the same in each FL round, and the initialization training time for the auxiliary models increases to $c \times n$. However, we have already explained in our response to weakness 2 that this situation is almost impossible.
> >
> > For weakness 5, as far as we know, the "Local Data Distribution Reconstruction Scheme" is not common in replay based continuous learning and federated continuous learning. The common replay scheme is to use the auxiliary model to generate an equal amount of replay data for each batch of real data, and use old task models to determine the type of replay data (see ablation experiment section, lines 1030-1044). The premise of our 'local data distribution reconstruction scheme' is that the auxiliary model can generate data for each category in a controllable manner, client can create a replay dataset before local training begins rather than generating an equal amount of data for each batch, greatly reducing the calculation cost and minimizing the errors caused by generate replay. This is difficult to achieve with existing FCL methods that use a single global auxiliary model. Secondly, the main problem with existing FCL methods is that the convergence rate of the model will decrease under the FL setting of high data heterogeneity, meaning that they will face the problem of poor performance caused by the non convergence of the task model and auxiliary model in the early and middle stages of FL training (see lines 081-093, lines 320-322 and Appendix E). Combining these methods with our local data distribution replay scheme also fails to effectively reduce the local replay errors.
> >
> > For weakness 6, there may be disagreement in the understanding of 'scalable'. The 'scalable' of our method is manifested in two aspects: 1. Our method is scalable for the situations where categories gradually increase, we only need to train new auxiliary sub models for the newly added categories. However, other existing FCL schemes may face the problem of insufficient capacity of auxiliary, even if the capacity is sufficient, it will require much more training time to adjust its auxiliary model which is much larger than our auxiliary sub model. We demonstrated this in the third baseline experiment (see lines 486-507, Appendix E.3) that our method performs much better than other FCL methods as the data categories increase with FL rounds. 2. Our method can be extended to other application areas, taking the time-series data generation as an example, we only need to use the encoder and decoder as both the task model and auxiliary model for each category, the specific work is already being prepared for submission, and we will not disclose the details here. But most of the other FCL methods are designed for the image field. For communication, the client of our method needs to send local task models and auxiliary sub models updated on real data to the server at each FL round to update the corresponding cache. For the auxiliary sub models of other categories, server can use the cache from the previous FL round (see lines 385-387). Other FCL methods also require the transmission of auxiliary models, ensuring that our method’s communication load is not significantly higher than other FCL methods under high data heterogeneity (such as baseline experiments). For computation costs, our additional computational cost compared to other FCL methods lies in the initialization training of each auxiliary sub model. We explained in our response to weakness 4 that how we can reduce the number of initialization training iterations.

---

> > > ### Author Response · Authors · 2024-11-20
> > >
> > > For question 1, "reduces communication bandwidth" and "real-time requirements" are aimed at the first line of "distributed machine learning". Traditional distributed learning methods require real-time transmission of raw data and gradients between different load nodes to achieve better learning results. Regarding the "reduces communication bandwidth", since that each parameter of the model corresponds to a gradient term, the communication consumption of transmitting gradient to updates the entire model is equivalent to the consumption of transmitting the entire model. However, the client of federated learning uploads local model with multiple local updates (i.e., the accumulation of local gradients) to the server, thereby reducing communication consumption. For "real-time requirements", poor network quality can directly affect the training progress of traditional distributed learning solutions while federated learning only requires model aggregation of clients that can connect to the server. Personalized federated learning methods are also more capable of solving the problem of slow convergence caused by asynchronous FL training that may occur in traditional federated learning methods in this situation.
> > >
> > > For question 2, there is no Y-axis in Figure 1, and the lower left corner of the right subgraph represents the starting point of virus mutation (i.e. COVID-19 BA. 2.86). Each branch represents a variant of the virus, and each colored dot on the branch indicates the time when a variant of the virus was discovered in a certain location in Europe. For details, please refer to the webpage link provided in the caption of Figure 1.
> > >
> > > For question 3, unknown data heterogeneity means that the differences in data distribution between different clients may be large or small, making it difficult to train a single global model that performs well on all clients. Therefore, personalized global model with good performance and strong generalization ability should be customized separately for each client. We have explained this in the previous sentence (see lines 044 and 054) of this sentence.
> > >
> > > For question 4, the optimization problem F1 (i.e. Eq1) is a summary of our global objective for FL and PFL, which is to minimize the task driven loss of the global model on each client. The only difference is whether the model sent by the server to the client is a global task model or a personalized global task model (see lines 186-191).
> > >
> > > For question 5, it raises the key to our design of category decoupled auxiliary model architecture. Due to the decoupling design of categories, our auxiliary model can effectively remember the historical distribution of data on each category. Specifically, due to the heterogeneity of data statistics in common situations mostly reflected in categories (see lines 283-284), we choose to train the auxiliary sub model of each category with a small amount of transfer learning on newly encountered real data of that category then make the replayed distribution nearing the mean distribution of all known data features in that category. When there is no real data for a category in each FL round, the auxiliary sub model corresponding to that category is not trained (see lines 324-329, lines 364-374), so there is no need to consider the catastrophic forgetting caused by data of other categories on the auxiliary sub model. For the server, the latest updates of each category's auxiliary sub model uploaded by each client are cached, so clients only need to upload the auxiliary sub models updated in each FL round to the server.
> > >
> > > For question 6, it involves the problem of model poisoning in FL security, where some clients attempt to affect the model performance of other clients during global model aggregation by incorporating local task models with error information. In our framework, the personalized global task model obtained by each client is aggregated on the approximation of the local distribution replayed by its auxiliary model (see lines 385-395), meaning that the aggregation weight of the local task models with error information will be reduced due to their poor performance in the replay data distributions of other clients, making it difficult to affect the performance of personalized global task models of other clients. For the local training phase, since the auxiliary models on other clients are not contaminated by erroneous information, other clients can correct the poisoning information contained in the global task model with knowledge transfer (see lines 345-363). We discussed this issue in the robustness section of the appendix (see lines 1572-1576)

---

> > > > ### Author Response · Authors · 2024-11-20
> > > >
> > > > For question 7, to our knowledge, existing CL and FCL methods based on generate replay tend to achieve good results through various loss functions (see our response to weakness 5). The idea of the "Local Data Distribution Reconstruction Scheme" is closer to the CL methods that building a core set on real data, but the actual contents of those methods are not similar to our method and cannot be used in the situations of "restricted storage" (see lines 035-037).
> > > >
> > > > For question 8, the FL setting of all clients may not participate in every FL round is called asynchronous FL, this setting only affects the global information contained in the global task model and may reduce the generalization ability of personalized global models on unknown data but have little impact on the performance of personalized global model on the client side. We discussed this issue in the robustness section of the appendix (see lines 1581-1585).
> > > >
> > > > For question 9, taking the CIFAR100 dataset containing 100 categories as an example, each client randomly divides $100$ categories into $100/2=50$ tasks with non overlapping categories before FL training begins, so the task combinations for different clients are different. Afterwards, each client executes a task belonging to that client in each FL round then tests the personalized global task model on the data categories corresponding to all previous tasks of that client. After 50 FL rounds, all tasks on each client will complete, and the FL training end. The details of the changes of the training data and testing data are similar to the other two baseline experiments.
> > > >
> > > > For question 10, let's give an example based on weather data: assuming that the data distribution of rainy and snowy is different, when the temperature in a region is above zero degrees, its weather changes between sunny and rainy. After a rapid cooling, the temperature remains below zero degrees, and its weather changes between sunny and snowy, that is, the data distribution of rainy is replaced by snowy.

---

> > > > > ### Comment · Reviewer_RaS3 · 2024-11-22
> > > > > **Thank you for the answers**
> > > > >
> > > > > I appreciate the thorough answers. Thank you, authors.
> > > > >
> > > > > **Referring to the reply to question 7 and weakness 5**: The "restricted storage" constraint of your work is quite similar to the whole motivation of continual learning, so I am not sure why the CL-based data reconstruction scheme "cannot be used in the situations of restricted storage".
> > > > >
> > > > > The authors have mentioned that "Local Data Distribution Reconstruction Scheme is not common in replay-based CL ...". To clarify, does it mean that there are at least some works in the area of continual learning which can generate a replay dataset before local training begins?
> > > > > If the "Local Data Distribution Reconstruction Scheme" can generate data before seeing the upcoming streaming data, shouldn't the reply-based methods which "generates an equal amount of replay data for each batch of real data" outperform the proposed scheme?
> > > > >
> > > > > **Referring to the reply to question 8**: Asynchronous learning is when different clients' trained models are gathered at different rounds (or at different times), rather than the server waiting for all the updates to arrive before aggregating the models and starting a new round (see Figure 4 in [1]). What I was talking about is cross-device FL. I am curious to, are there any results on clients dropping out in the middle of a round or partial participation for all rounds?
> > > > >
> > > > > [1] Papaya: Practical, Private, and Scalable Federated Learning, Huba et al. (MLSys 2022, https://proceedings.mlsys.org/paper_files/paper/2022/file/a8bc4cb14a20f20d1f96188bd61eec87-Paper.pdf)
> > > > >
> > > > > **Referring to the reply to weakness 6**: While I understand that the definition of scalability is different between the authors and I, I was thinking more in terms of "if the storage is restricted for the data, do the clients have capacity to even store the following 3 each round: (a) A set of auxiliary models, (b) a personalized model specific to a client, and (c) an aggregated global model?"
> > > > >
> > > > > I still believe that the authors should have tables or equivalent form of data presentation for describing communication and computation costs of all the baselines and their proposed methods. It's difficult to visualize the theoretical costs.
> > > > >
> > > > > **Referring to the reply to weakness 4**: I still believe that just claiming "we protect privacy because our model is trained in so and so manner" is weaker than showing empirical results on some membership-inference or equivalent attacks. If you need differential privacy with your method, in order to preserve privacy, then claiming "improving model generalization ability while protecting privacy" (line 96) for the current method is still extravagant.
> > > > > ___
> > > > > Overall, I would be keeping my score for now due to the following reasons: (a) Readability, and (b) Practicality (computation and communication-wise). Furthermore, I am still uncertain about the novelty of "Local Data Distribution Reconstruction Scheme".

---

### Official Review · Reviewer_miPs · 2024-11-03

**Soundness:** 2
**Presentation:** 2
**Contribution:** 2
**Rating:** 3
**Confidence:** 4

**Summary:**

This paper discusses personalised federated learning (pFL) on flowing Data Heterogeneity under Restricted Storage. Compared to existing pFL framework, the proposed solution, i.e., pFedGRP redesigns the generated replay scheme so that is can mitigate catastrophic forgetting and improve model generalization ability. Specifically, this paper applies an idea of category decoupling for adapting continuous learning into pFL to achieve knowledge transfer and personalized aggregation in pFL.

**Strengths:**

This paper somehow extends the pFL setting. It proposes a new pFL framework where the CL has been introduced with restricted by the category decoupling. Experimental results have demonstrated the effectiveness of the proposed method.

**Weaknesses:**

The motivation of the proposed method is not very clear. The methodology is not well presented. Especially, why this proposed pFedGRP is more effective than existing methods, which aspect, is less clear to me. Please refer to the question section for details.

**Questions:**

This paper highlights the Restricted Storage. However, it is not explicitly explained what case refers to "restricted storage" till the experiment section. "Due to the FL setting of Flowing Data Heterogeneity under Restricted Storage where the client is unable to access the real data encountered in the previous task, each client can build up to 150 tasks on the MNIST and FashionMNIST datasets and up to 125 tasks on the Cifar10 dataset."---I don't understand why this can be considered as the experiment of "flowing Data Heterogeneity under Restricted Storage".

Why the setting has to be "flowing Data Heterogeneity under Restricted Storage"? Does it mean existing studies already addressed "flowing Data Heterogeneity"? This aspect is not fully reviewed which results in my concerns on the motivation and novelty of this paper.

This paper claims that the proposed method can "achieve the goals of personalized aggregation, mitigating catastrophic forgetting and improving model generalization ability while protecting privacy." However, the experimental result does not support this claim as pFedGRP performs poor in forgetting. This somehow indicates the good performance on accuracy is not because of the proposed mechanism but the experimental setting of  "Restricted Storage" as in this case, there is no need of memorizing.

---

> ### Author Response · Authors · 2024-11-20
>
> Thank you very much for taking the time to read our work and providing suggestions and questions! Below are our responses to the various weaknesses and questions you have raised:
>
> For question 1, regarding "restricted storage" in the baseline experiment, we set each client's task to contain two categories of data, with $200$ data for each category, resulting in 400 data for each task (see section 5.1). Due to the training data of the MNIST and FashionMNIST datasets being $60000$, they can be divided into $60000/400=150$ non overlapping training data parts. The training data of Cifar10 dataset is $50000$, so it can be divided into $50000/400=125$ non overlapping training data parts. The partitioning of the test dataset is similar to the partitioning of the training dataset. Under our setup, even if the categories of the task in current FL round to the same of the categories of the task in previous FL round, the accessable training data parts in different FL round are still different, that is, the client cannot access the data encountered in the previous task. However, the data on the local test data set of the client is cumulative, meaning that the model not only needs to be tested on the test data part corresponding to the current task, but also on the test data parts of all previous tasks. Regarding "Flowing Data Heterogeneity", taking the baseline experiment section of the above dataset as an example (lines 406-419), we set each client to be randomly divided into $10/2=5$ tasks with non overlapping categories before FL training, meaning that different clients are likely to have different data categories in the same FL round. After the start of FL training, the tasks of each client will switch between adjacent FL rounds, meaning that the categories of data that each client can access in two adjacent FL rounds are different and do not overlap. Therefore, the data heterogeneity between clients will also change with the change of FL rounds (see Appendix A).
>
> For question 2, the existing research in the FL field to address issues similar to "heterogeneity of flowing data under restricted storage" mainly focuses on Federated Continuous Learning (FCL) (see lines 079-093) with relevant work in the past two years presented in lines 145-166. These methods mainly focus on training a single global model for all clients (see optimization problem F2, line 217) and face the problem of slow convergence speed of the model (whether task model or auxiliary model) in high data heterogeneity, resulting in poor performance in the early and middle stages of FL training (see Appendix E.1, E.2). In the real world, the categories of data constantly arriving at the client is unknown. If the client continues to receive data of new categories, existing FCL methods may face difficulties in converging (see Appendix E.3). To address this issue, we propose pFedGRP inspired by personalized federated learning to solve the problems above.
>
> For question 3, the forgetting metric (FM) represents the stability of a method's performance throughout the entire FL training process, and is only used to measure the advantages and disadvantages of methods with similar average accuracy (AA) (see lines 429-431 and Appendix C.2). It should not be used to compare methods with large differences in average accuracy. Considering an extreme case where the accuracy of a method remains constant at $0$ throughout the FL training process, and its forgetting index is $0$, is this method performing well? Then, we provide a comparative experiment in Appendix D.2 to increase the number of overlap categories between adjacent tasks from $0$ to $6$. The increase in the number of overlap categories not only reduces the degree of variation in local data distribution, but also relaxes the experimental settings for "restricted storage". For example, when the number of overlap categories is $6$, each task contains data from $8$ categories, and 75\% of the data encountered by adjacent tasks is of the same category.

---

### Official Review · Reviewer_MNfL · 2024-11-04

**Soundness:** 2
**Presentation:** 2
**Contribution:** 2
**Rating:** 5
**Confidence:** 3

**Summary:**

The paper introduces pFedGRP, a framework for pFL aimed at handling Flowing Data Heterogeneity under Restricted Storage. This setup reflects real-world scenarios where data distributions shift over time on each client, and clients have limited data storage. To address these challenges, the framework uses a category decoupled data generator and a local data distribution reconstruction scheme. This approach reduces errors in data replay and improves knowledge transfer between clients.

**Strengths:**

1. The paper tackles the novel problem of personalized federated learning under flowing data heterogeneity with restricted storage, which closely mimics real-world scenarios where data distributions shift over time, and clients are limited in storage capacity. The formulation of Flowing Data Heterogeneity under Restricted Storage is an innovative extension of traditional pFL, making the problem more applicable to diverse, real-world environments where federated learning is deployed.
2. The experimental design demonstrates quality through comprehensive testing across multiple datasets, including MNIST, FashionMNIST, CIFAR-10, CIFAR-100, and EMNIST. These experiments reflect a thorough analysis of accuracy and robustness metrics, especially focusing on average accuracy (AA) and average forgetting measure (AFM) to evaluate both model performance and resilience to catastrophic forgetting. The comparisons include eight baseline methods spanning standard federated learning (FL), pFL, and FCL methods, providing a robust benchmark for assessing pFedGRP’s effectiveness.
3. The paper is generally well-written and organized. The introduction and related work sections give clear motivations and delineate the scope of pFL under dynamic data.

**Weaknesses:**

1. While the auxiliary model provides a solution for replay-based continual learning, its computational cost and storage implications for each client could be substantial. The paper would benefit from a deeper discussion of resource constraints, especially in low-power devices or large-scale networks with many clients. Additionally, a performance evaluation showing the computational overhead for maintaining the auxiliary model on each client would offer a clearer picture of its feasibility.
2. The success of pFedGRP heavily depends on the accuracy of generated replay data. Any inaccuracies in replay could misrepresent past distributions, thus introducing bias or increasing the error in training. While the paper discusses mitigating catastrophic forgetting, more insight into replay quality control mechanisms could enhance robustness. This could involve ablations showing how variations in replay accuracy affect model performance, or exploring regularization methods to enforce consistency between real and replayed data distributions.
3. The methodology section is dense, especially the part on personalized aggregation and knowledge transfer via auxiliary models. While it is generally clear, some steps (e.g., the loss function design in personalized aggregation) could use further elaboration to improve understanding. Readers less familiar with pFL and FCL may find it challenging to grasp these nuanced details without additional guidance. Consider adding a step-by-step breakdown in the appendix to enhance clarity.

**Questions:**

1. Could you provide further details on the replay quality? For example, how do you handle potential discrepancies between the generated replay data and the actual historical data distribution? Is there a mechanism for adjusting the auxiliary model if replay quality degrades over time?
2. What is the typical storage and computational cost of maintaining the auxiliary model on each client? Have you considered any optimization techniques to reduce this burden, especially for resource-constrained devices?
3. Given that the framework is tailored for dynamic data, could you conduct an ablation on the sensitivity to data drift? This might involve varying the rate of data distribution change to observe its impact on accuracy and forgetting metrics.

---

> ### Author Response · Authors · 2024-11-20
>
> Thank you very much for taking the time to read our work and providing many suggestions and questions! Below are our responses to the various weaknesses and questions you have raised:
>
> For weakness 1, there are two goals for continuous learning based on generated replay: reducing the waste of storage caused by a large amount of duplicate or similar data, and avoiding the restrictions of data regulations on the transmission and storage time of real data (see lines 035-037). Most existing federated continuous learning (FCL) works uses a single large-scale auxiliary model to memorize all known data distributions and alleviate catastrophic forgetting during training by generate replay (Chapter 2.2),  which not only reducing the effectiveness of generating replays (see lines 079-093) but also bringing additional cost on memory and computing power requirements:
> Firstly, the existing methods use auxiliary models to generate equal amounts of replay data for each batch of real data during local training and use old task models to judge the category of replay data (see ablation experiment section, lines 1030-1044), which means that these auxiliary models and generated replay data need to be stored in memory (otherwise the training speed will be reduced due to IO operations), the amount of generated data is equivalent to the product of the number of epochs and the total number of real data. Our method effectively improves the quality of generated replays and just needs to generate the data before the local training starts based on our local data distribution reconstruction scheme, the quantity of generated data in the worst-case setting is the product of the maximum number of categories in historical data and the maximum number of each category in real data. General speaking, it is only equivalent to the number of generated data by existing methods in the first few epochs, and these generated data can be placed on the hard disk and combined with real data to form a batch waiting to be called into memory.
> Secondly, the category decoupled auxiliary model architecture proposed in our scheme is based on the assumption that the statistical heterogeneity of data is mostly reflected in categories (see lines 283-284). The auxiliary sub models corresponding to each category only need to learn the data distribution of a single category, which means that the problem of catastrophic forgetting can be alleviated during training, and only a small amount of transfer learning is needed to fit the latest distribution of data in that category. When there is no real data for a category, the training of the auxiliary sub model corresponding to that category is not performed (see lines 324-329, lines 364-374). However, existing methods adopt a single auxiliary model structure, which means that they require a much larger number of parameters than our auxiliary sub model to fit the data distribution of all categories, and they also need to solve their catastrophic forgetting problem based on generative replay during training (such as FedCIL, AF-FCL, see Appendix B). Taking FedCIL as an example, it uses ACGAN as both auxiliary model and task model. The default epochs for each client’s local training is 400. During local training, it uses global model and the previous round's local model to generating replay to alleviating catastrophic forgetting. After global aggregation, it still requires 100 epochs for fine-tuning in server. Despite such a large computational demand, it is still difficult to quickly fit complex data distributions (see Appendix E).

---

> > ### Author Response · Authors · 2024-11-20
> >
> > For weakness 2, we adopt the category decoupled auxiliary model architecture to control the number of replayed data for each category while ensuring a small error in the distribution of replayed data on each category, thereby achieving the goal of personalized aggregation and alleviating catastrophic forgetting during local training. We provided a comparative experiment (see lines 1030-1044) in the ablation experiment section, where we replaced our replay scheme with other FCL methods’ replay scheme. The results showed that our replay scheme outperformed the above methods in both accuracy and robustness.
> >
> > For weakness 3, we have provided detailed steps of our method in the pseudocode (see Appendix C.1).
> >
> > For question 1, we use local distribution reconstruction scheme to reduce the number of replay data during local training to avoid introducing more replay errors (see lines 283-317). For other FCL methods that use a single global auxiliary model, the categories of the generated data are uncontrollable, so our local distribution reconstruction scheme cannot be used in those methods. When the replay quality degrades over time, it means that the data distribution of a single category is too complex for the auxiliary sub model to fit the data distribution of that category. It should be considered to increase the parameter count of the sub auxiliary model or replace the architecture of the auxiliary sub model. This also means that other FCL methods which use a single auxiliary model face the problem of insufficient model capacity in the current situation, resulting in poorer performance (see lines 318-323).
> >
> > For question 2, we describe the worst-case situation here: Assuming we use the auxiliary sub model which has the same scale as other baseline methods' auxiliary model, the number of epochs for local training of the auxiliary model is $n$, the amount of local data in each task is $m$. Since our method only trains each auxiliary sub model on the corresponding real data of the category without generating replay, the total amount of data that needs to be traversed for training is $m \times n$. For other methods, as they typically rely on equally generated replays to alleviate catastrophic forgetting, the total amount of data that needs to be traversed for training is $2 \times m \times n$. However, in practice, since each auxiliary sub model only needs to fit the data distribution of a single class, the number of training epochs is much smaller than other methods that needs to fit the data distributions of all known classes. For the storage space, assuming the number of categories encountered by each client is $c$, the worst-case sitting for our auxiliary model to occupy storage space is $m \times c$, for other methods is m. However, each auxiliary sub model does not require such a large model size to fit the data distribution of a single category.
> >
> > For question 3, we provide a comparative experiment in Appendix D.2 to increase the number of overlap categories between adjacent tasks from 0 to 6. Increasing the number of overlap categories means reducing the rate of local data distribution changes.

---

### Note · Authors · 2024-11-25

I have read and agree with the venue's withdrawal policy on behalf of myself and my co-authors.